# Adaptive Vision Encoders: Balancing Efficiency and Robustness in Vision-Language Models

## Abstract

Vision-language models (VLMs) demonstrate impressive capabilities in visual question answering and image captioning, acting as a crucial link between visual and language modalities. However, existing open-source VLMs rely heavily on pretrained vision encoders, such as CLIP. Despite CLIP's robustness across diverse domains, it still exhibits significant image understanding errors. These errors propagate to the VLM responses, resulting in sub-optimal performance. In our work, we propose an efficient and robust method for updating vision encoders within VLMs. Our approach selectively and locally updates the model parameters, leading to substantial performance improvements on data where previous mistakes occurred, while maintaining overall robustness. We demonstrate the effectiveness of our method during offline and continual few-shot updates, simulating a model editing regime for VLMs. While our method also scales efficiently and effectively to adapting the language model (LLM) component of the VLM, we show that separately updating the vision encoder can be a very efficient alternative. This approach improves VLM performance with less than 10x the compute resources required for updating the LLM. Our method is also supported by theoretical justifications on the parameter selection strategy.

## 1 Introduction

Large Language Models (LLMs) have transformed the landscape of natural language understanding and generation, revolutionizing a wide range of domains and applications. These advancements bring us one step closer to creating useful and reliable automated assistants. Given that vision and visual understanding play a crucial role in intelligent agents expected to operate in the real world, Vision Language Models (VLMs) have emerged. These models either incorporate embeddings from vision-only models or are trained end-to-end with both vision and language input. Remarkably, VLMs consistently achieve impressive performance across question-answering and image-captioning benchmarks. We refer to Ghosh et al. (2024) for a recent survey on VLMs. Approaches that rely on pretrained vision encoders typically use variants of the CLIP model, which is kept frozen in the vision-language binding process. CLIP (Radford et al., 2021), a widely deployed vision and text transformer, stands out for its robustness to domain shifts and outstanding capabilities of recognizing a large range of objects, scenes and actions. However, our evaluation reveals specific limitations in CLIP's performance. Specifically, when tested on an action recognition dataset featuring various simple actions with moderate image quality, CLIP exhibits substandard performance and seems easily confounded by the image content. Other works Liu et al. (2024b); Zhu et al. (2023); Chen et al. (2023); Li et al. (2023a) reveal similar shortcomings of CLIP for particular use cases. These findings underscore weaknesses in visual understanding of CLIP, specially on challenging and previously unseen domains, and prompts the need for continuous model improvements to address these imperfections.

In order to enable VLMs to adapt to new data or domains, we envision a realistic scenario where the model can be updated efficiently with minimal computational resources while maintaining its strong performance on other data and domains. In other words, we aim to correct mistakes effectively while preserving existing knowledge.

Figure 1: Samples from TSI dataset (bottom) compared to DALL-E generated images (top) with labels indicated above. Right: LLaVA's correct response to a DALL-E image versus a wrong response to a TSI image of the same label (cutting food).

Given the composite nature of VLMs, which combine vision encoders and language models, the crucial question arises: Which components are better suited for targeted updates? To address this, we conducted separate fine-tuning experiments on the vision encoder and the Language Model (LLM) using a dataset where the VLM exhibited numerous mistakes. The results were intriguing: *separately* updating the vision encoder significantly improved the performance on the specific data of interest achieving even better accuracy than updating the LLM. Updating the vision encoder is more efficient as it contains far fewer parameters than the language model and can improve the entire family of VLMs that are build upon it. Our findings suggest that separately updating the vision encoder provides a more robust alternative to LLM updates when visual shift is the primary source of errors.

Despite the effectiveness and efficiency of vision encoder adaptation, continuous and frequent updates can lead to performance deterioration. Therefore, we recognize the need for not only efficient updates, but also the localization of parameter updates to the data at hand, in order to limit degradation in unrelated areas of knowledge. This means that adapting the model should not change all the parameters uniformly but rather localize and limit the update to a small subset of parameters. This approach helps preserve as much of the previously embedded knowledge as possible. Parameter-efficient fine-tuning methods often degrade unrelated knowledge similarly to full fine-tuning, as shown by Zhang et al. (2024) and demonstrated in our experiments. For instance, LoRA's (Hu et al., 2021) low-rank updates still alter all model parameters, which can result in performance decline upon multiple updates.

To achieve localized updates, we propose modifying only task-relevant parameters while keeping the rest intact. This approach aligns with Language Model Editing Meng et al. (2022), though existing methods are typically specific to factual updates and not easily applicable to vision-related updates. We identify which parameters to update by masking those that preserve the gradient norm of the model's estimated update, selecting parameters with the greatest gradient norm. For MLP layers, we follow SPU (Zhang et al., 2024) by selecting the top k parameters based on gradient norm. Our method generalizes to attention heads, selecting specific heads by the same rule. We combine these masks with low-rank updates (Hu et al., 2021), achieving both locality and efficiency.

We validate our method across various benchmarks, both by updating CLIP and by enhancing VLM models based on CLIP. Our approach demonstrates superior performance and preserves the model's generic knowledge. While our focus lies on updating the vision encoder, our method is generic and applicable to any transformer model whether for vision, language, or any other modality. Our contribution are as follows: 1) We evaluated CLIP on out-of-distribution benchmarks and observed shortcomings in certain scenarios. These limitations are then propagated to the VLMs that leverage CLIP's embeddings. 2) Our work demonstrates that updating the vision encoder *separately*, specifically on data where CLIP fails, can significantly correct VLM mistakes on previously unseen images from this data. 3) We propose a novel parameter-efficient tuning method LoRSU that not only targets efficiency but also ensures the preservation of the model's generic knowledge. We evaluate our approach on offline adaptation as well as the challenging continual few-shot adaptation. We compare adapting the vision encoder separately to adapting the LLM with our method compare to LoRA (Hu et al., 2021) on the LLM. 4) We show state of the art results and robustness both when adapting the vision encoder separately and when adapting the LLM. Adapting the vision encoder can

be more than 10x faster than adapting the large language model. To the best of our knowledge, we are the first to show that adapting the vision encoder separately can be a very efficient and effective approach for improving the VQA performance on downstream tasks.

We validate our method across various benchmarks by updating CLIP and enhancing VLM models based on CLIP. Our approach demonstrates superior performance while preserving the model's generic knowledge. Although our focus is on updating the vision encoder, our method is generic and applicable to any transformer model, whether for vision, language, or other modalities. Our contributions are as follows: 1) We evaluated CLIP on out-of-distribution benchmarks and identified shortcomings in certain scenarios. These limitations are then propagated to the VLMs that leverage CLIP's embeddings. 2) Our work demonstrates that updating the vision encoder *separately*, specifically on data where CLIP fails, can significantly correct VLM mistakes on previously unseen images from this data. This approach proves to be more robust against catastrophic forgetting of the model's generic knowledge compared to updating the language model. 3) We propose a novel parameter-efficient tuning method, LoRSU, that not only targets efficiency but also ensures the preservation of the model's generic knowledge. We evaluate our approach on offline adaptation as well as the challenging continual few-shot adaptation. We compare adapting the vision encoder separately to adapting the LLM with our method and to LoRA (Hu et al., 2021) on the LLM. 4) We show state-of-the-art results and robustness both when adapting the vision encoder separately and when adapting the LLM. Adapting the vision encoder can be more than 10x faster than adapting the large language model. To the best of our knowledge, we are the first to show that adapting the vision encoder separately can be a very efficient and effective approach for improving VQA performance on downstream tasks.

In the following we discuss closely related work in Section 2 and then showcase how CLIP weaknesses are manifested in the VLM VQA responses in Section 3. We present our approach in Section 4 and validate empirically our claims in Section 5. We conclude in Section 6.

## 2 RELATED WORK

Large language model and vision language models are strong foundation models but are still prone to mistakes and their knowledge can get outdated, consequently it is important to develop efficient updates that preserve unrelated knowledge. The main line of work in this area focuses on LLM editing, where previous factual knowledge has changed and the model must be updated effectively on those changes. Most notably Meng et al. (2022) and Ilharco et al. (2022) first analyze the models to identify specific layers for editing, i.e., where factual knowledge are "stored" in the model, and then apply algebra-based or meta-learning methods to adjust the weights of these localized layers. To insure the locality of the updates these methods usually leverage additional sets of parameters representing unrelated factual knowledge.

Another line of work focus on updating the model for a new task or dataset with parameter efficient finetuning. Low Rank Updates (LoRA) (Hu et al., 2021) approximates the parameter updates by a low rank matrix, achieving similar performance on the target task by optimizing only 1% of the parameters compared to the full model. The original version of LoRA updated only attention layers. Subsequently, several extensions have been proposed to enhance LoRA which modify all layers. Various options are available including adapting the learning rate (Hayou et al., 2024) of the low rank matrix, using an adaptive rank (Zhang et al., 2023) or decomposing the update matrix into magnitude and direction Liu et al. (2024c). These approaches focus solely on efficiently updating the network without considering the impact on model performance for other unrelated tasks or enforcing any locality to specific layers or parameters. It is worth noting that LoRA drop (Zhou et al., 2024) attempts to localize the updates to specific layers. It initially allows a few iterations of LoRA updates and then assesses the impact of each low-rank update on individual layers and selectively updates only those layers where the change exceeds a specified threshold. However, this selectivity remains at the layer level and depends on the change introduced by a few full updates. In contrast, we treat each layer differently based on its structure and assess the relevance of individual parameters to the task at hand. We then holistically combine the importance and relevance of these parameters with low-rank updates.

In the context of updating vision models for specific tasks, SPT (He et al., 2023) estimates a mask of updates based on parameter sensitivity to the task. Depending on the number of relevant

| Clip-L-14 | | |
|:---:|:---:|:---:|
| **ImageNet** | **TSI** | **DALLE** |
| 76.6 | 13.2 | 90.9 |

Table 1: Clip-L-14 zero-shot Accuracy (%) on ImageNet, TSI and DALLE datasets. TSI accuracy is much lower than DALLE.

| LLaVA 1.5 | | |
|:---|:---:|:---:|
| Method | **DALLE** | **TSI** |
| **Zr-Shot** | 91.1 | 53.1 |
| Adaptation of **LLama-2+Pj** | 88.5 | 73.3 |
| Adaptation of **CLIP-L-14** | **91.1** | **75.5** |

Table 2: VQA Accuracy (%) comparing Zeroshot to fine-tuning the LLM (with LoRA $r = 8$) and the MLP projector as well as to fine-tuning CLIP-L-14 *separately*.

parameters, either low-rank or sparse updates are performed (using a threshold). With regards to continual updating CLIP while maintaining its generalization performance and reducing forgetting, SPU (Zhang et al., 2024) treats layers of the transformers differently, and inspired by knowledge neuron theory, SPU localizes the updates to the first feedforward layer of each transformer block and then only relevant parameters to the task at hand are updated. We further refer to De Lange et al. (2021) for a survey on continual learning. In our approach, we select and identify relevant parameters to the current data. However, we generalize the updates to all layers while preserving the specificity of each layer. We choose masks that maintain the gradient norm of parameter updates and combine them with LoRA on selected attention heads, striking a balance between adaptivity and stability.

## 3 DO WEAKNESSES IN CLIP PROPAGATE TO THE VLM?

VLMs are either trained end-to-end or as composites of separate vision and language models. In the latter, the vision encoder typically remains frozen during VLM training. CLIP (Radford et al., 2021), a vision transformer model trained by contrasting vision and language, is the primary vision encoder used in this line of VLMs. It is employed in MiniGPT Zhu et al. (2023), MiniGPTv2 Chen et al. (2023), BLIP2 (Li et al., 2023a), CogVLM (Wang et al., 2023), Kosmos-2 (Peng et al., 2023), LLaVA (Liu et al., 2024b), and LLavaNext (Liu et al., 2024a). CLIP, trained on vast vision and language pairs, shows unprecedented robustness to domain shifts, including adversarial scenarios (Mayilvahanan et al., 2023). However, Mayilvahanan et al. (2023) noted that CLIP's large pretraining dataset might include examples from out-of-distribution benchmarks. Our objective is to examine how CLIP's possible failure modes affect VLM behavior, which we leverage to design an efficient method for adapting VLMs with pretrained vision encoders.

For that purpose, we opt for a simple yet realistic evaluation. We considered the Toyota Smart Home (TSI) dataset (Das et al., 2019), a dataset of daily living activities staged in a home-like environment. This dataset cannot be publicly crawled from the web, it is only accessible upon request. This makes it unlikely to have been used to train CLIP. Further, the data depict elderly people activities (age bias), blurred faces (blurring effect) and is captured from a mounted camera with somewhat low resolution, yet the actions are easily recognizable to the human eye. For more details, refer to the Appendix and Experiments Section 5. Interestingly, when evaluating CLIP on images from the TSI dataset, we observed only moderate performance and encountered numerous mistakes. Table 1 reports CLIP accuracy on TSI dataset compared to ImageNet accuracy. Now, we examine the responses of a VLM using CLIP's vision encoder, namely LLaVA 1.5 (Liu et al., 2024b). Our test revealed similarly poor performance as shown in Table 2. We refer to Figure 1 for an example response and to the Appendix for more examples on TSI images. The main failure modes of LLaVA 1.5 on TSI are hallucination of wrong activities or describing the background rather than the action.

Further to isolate whether this suboptimal performance is a result of CLIP's limited knowledge of the performed activities or its lack of robustness to the distribution shift present in the images, we leveraged diffusion models, specifically DALL·E 2, to generate images of people performing the same actions. After verifying the quality of these generated images, we tested CLIP's predictions on them. Remarkably, CLIP accurately recognized the actions on the synthetic images. Similarly, LLaVA also provided very accurate descriptions of the generated images. Figure 1 show some generated images compared to images from TSI (Das et al., 2019) of similar activities.

This case study shows how CLIP weaknesses on new domain propagate to the full VLM and are manifested in the visual question answering performance. Next, we address the question of *how to efficiently update both the vision encoder and the corresponding VLM while maintaining the overall robustness and generalization of both components.*

## 4 LOW-RANK ADAPTATION WITH STRUCTURED UPDATES

To address the challenge of efficiently fine-tuning large-scale visual encoders and transformer-based models, including LLMs, without causing catastrophic forgetting (i.e., degradation in performance on previously learned tasks), we propose a novel parameter-efficient fine-tuning method called *Low-Rank Adaptation with Structured Updates* (**LoRSU**).

LoRSU updates specific parameters within each transformer block in a resource-efficient manner, mitigating the risk of generic knowledge loss when fine-tuning for new tasks. Specifically, we selectively update a subset of parameters from the first linear layer in the MLP block of each transformer layer, as proposed in Zhang et al. (2024). While this approach reduces the fine-tuning burden, it may limit model flexibility as the remaining parameters in the transformer block remain fixed. To enhance flexibility, we further update the most informative attention heads based on the gradient of the task-specific loss.

More specifically, let a dataset $\mathcal{D}_t = \{\mathbf{x}_n, \mathbf{y}_n\}_{n=1}^{N_t}$ for the current task $t$ where $\mathbf{x}_n$ is an image with text description $\mathbf{y}_n$ and $\mathcal{L}(\boldsymbol{\theta}; \mathcal{D}_t) := \mathcal{L}_t(\boldsymbol{\theta})$ is the loss used for pretraining the transformer model and $\boldsymbol{\theta} \in \mathbb{R}^d$ is the full set of model's parameters. The standard Multi-head Self-Attention Mechanism (Vaswani et al., 2017), comprised of $H$ $D_h$-dimensional heads, is defined as the concatenation of multiple self-attention (SA) blocks:

$$\mathbf{q}^{(i)} = W_q^{(i)} Z^\top, \mathbf{k}^{(i)} = W_k^{(i)} Z^\top, \mathbf{v}^{(i)} = W_v^{(i)} Z^\top \in \mathbb{R}^{D_h \times N}, \tag{1}$$

$$A^{(i)} = \mathrm{softmax}(\mathbf{q}^{(i)\top} \mathbf{k}^{(i)} / \sqrt{D_h}) \in \mathbb{R}^{N \times N}, \tag{2}$$

$$\mathrm{SA}_i(Z) = A^{(i)} \mathbf{v}^{(i)\top} \in \mathbb{R}^{N \times D_h}, \quad i = 1, \ldots, H. \tag{3}$$

where $Z \in \mathbb{R}^{N \times D}$ is the input matrix of $N$ tokens of dimension $D$ and $W_q^{(i)}, W_k^{(i)}$, and $W_k^{(i)}$ are the query, key, and value matrices of learnable parameters for head $i$, respectively. The final MSA function is defined as

$$\mathrm{MSA}(Z) = \mathrm{Concat}\left[SA_1(Z), \ldots, SA_H(Z)\right] W_o \in \mathbb{R}^{N \times D}, \quad W_o \in \mathbb{R}^{HD_h \times D}, \tag{4}$$

Since we care to update the parameters of the heads that cause the largest changes in $\mathcal{L}_t(\boldsymbol{\theta})$, we compute the gradient of the loss with respect to the parameters of each head and then we update only those heads with the largest cumulative contribution to the loss change. Since the matrices $W_q^{(i)}, W_k^{(i)}, W_v^{(i)}$ are all the parameters of head $i$, we can define an importance score for each head by adding the squared values of their corresponding gradients $G_q^{(i)} = \nabla_{W_q^{(i)}} \mathcal{L}$, $G_k^{(i)} = \nabla_{W_q^{(i)}} \mathcal{L}$, $G_v^{(i)} = \nabla_{W_v^{(i)}} \mathcal{L}$, and $G_o^{(i)} = \nabla_{\widetilde{W}_o^{(i)}} \mathcal{L}$, i.e.

$$s_i = \sum_{m,l} \left( (G_q^{(i)}[m,l])^2 + (G_k^{(i)}[m,l])^2 + (G_v^{(i)}[m,l])^2 + (G_o^{(i)}[m,l])^2 \right). \tag{5}$$

We provide a theoretical justification of equation 5 in the next section. We update only the top-$k$ heads, based on their importance scores $\{s_1, \ldots, s_H\}$, $I \subset \{1, \ldots, H\}$, to be updated on the current task. Nevertheless, the number of parameters remain high due to the large weight matrices. Therefore, we parametrize the original weights using LoRA Hu et al. (2021) to further reduce the computational burden. The matrices $W_q^{(i)}, W_k^{(i)}, W_v^{(i)}, i \in I$ are now defined as

$$W_q^{(i)'} = W_q^{(i)} + A_q^{(i)} B_q^{(i)} \tag{6}$$

$$W_k^{(i)'} = W_k^{(i)} + A_k^{(i)} B_k^{(i)} \tag{7}$$

$$W_v^{(i)'} = W_v^{(i)} + A_v^{(i)} B_v^{(i)}. \tag{8}$$

Finally, to ensure that we only update $W_q^{(i)}, W_k^{(i)}, W_v^{(i)}, \forall i \in I$ we use a binary mask on the gradient vector with respect to all parameters of all attention heads. We keep the projection matrix $W_o$ frozen throughout optimization.

Regarding the first linear layer in the MLP module, $W_{\mathrm{fc1}} \in \mathbb{R}^{d \times D}$, we mask the gradients of $W_{\mathrm{fc1}}$ so only the most important parameters for the current task to be updated, i.e. we use the following biased gradient update.

$$\hat{\nabla}_{W_{\mathrm{fc1}}} \mathcal{L}_t = M_{\mathrm{fc1}} \odot \nabla_{W_{\mathrm{fc1}}} \mathcal{L}_t, \tag{9}$$

where $M_{\text{fc1}} \in \{0,1\}^{d \times D}$ is a zero-one mask that is built by choosing a proportion of the largest squared values of $\nabla_{W_{\text{fc1}}} \mathcal{L}_t$ in a similar manner as in Zhang et al. (2024) and $\odot$ is the Hadamard product.

### 4.1 THEORETICAL JUSTIFICATION

The importance scores in in equation 5 can be derived from the following constrained (binary) optimization problem

$$\mathbf{p}^* = \arg\max_{\mathbf{p} \in \{0,1\}^d} \frac{\|\mathbf{p} \odot \nabla_W \mathcal{L}(\boldsymbol{\theta}_0)\|^2}{\|\nabla_W \mathcal{L}(\boldsymbol{\theta}_0)\|^2}, \quad \text{s.t.} \quad \bigcup_{\ell=1}^{G} I_\ell \subset \{1, 2, \ldots, d\}, \quad \text{where } I_i \cap I_j = \emptyset, \ \forall i \neq j,$$

$$S = \sum_{\ell=1}^{G} s_\ell, \ \ s_\ell \leq |I_\ell| \ \forall \ell, \quad \|\mathbf{p}\|_0 \leq S, \tag{10}$$

Here $\boldsymbol{\theta}_0$ is the pretrained vector of parameters before we use the $\mathcal{D}_t$ for fine-tuning. The mask $\mathbf{p}^*$ is chosen so that the gradient norm of the masked gradients is as large as possible under the sparsity constraints.

**Definition 4.1.** The operator TOP-$S : \mathbb{R}^d \to \mathbb{R}^d$, for $1 \leq S \leq d$ is defined as

$$(\text{TOP-}S(\mathbf{x}))_{\pi(i)} := \left\{ \begin{array}{ll} x_{\pi(i)}, & i \leq S \\ 0, & \text{otherwise,} \end{array} \right.$$

where $\pi$ is a permutation of $\{1, 2, \ldots, d\}$ such that $|x_{\pi(i)}| \geq |x_{\pi(i+1)}|$, for $i = 1, \ldots, d-1$, i.e. the TOP-$S$ operator keeps only the $S$ largest elements of $\mathbf{x}$ in magnitude and truncates the rest to zero.

**Lemma 4.2.** For any $\mathbf{x} \in \mathbb{R}^d - \{\mathbf{0}\}$, $1 \leq S \leq d$, the optimal mask

$$\mathbf{p}^* = \arg\max_{\mathbf{p} \in \{0,1\}^d} \frac{\|\mathbf{p} \odot \mathbf{x}\|^2}{\|\mathbf{x}\|^2}, \ \ s.t. \ \|\mathbf{p}\|_0 \leq S,$$

has zeros everywhere except the $S$ largest elements of $\mathbf{x}$ in magnitude.

*Proof.* Rewriting the optimization problem as

$$\max_{\mathbf{p} \in \{0,1\}^d} \sum_{i=1}^{d} p_i x_i^2, \ \ \text{s.t.} \ \sum_{i=1}^{d} p_i \leq S,$$

we notice that this a trivial binary knapsack problem with maximum weight capacity $S$ and weights equal to one. Hence, the maximum is attained when we pick the top $S$ maximal $x_i^2$ elements. $\square$

*Remark* 4.3.

It holds that TOP-$S(\mathbf{x}) = \mathbf{p}^* \odot \mathbf{x}$.

**Corollary 4.4.** *The optimal mask $\mathbf{p}^*$ in equation 10 has zeros everywhere except for the indices $i \in \{j : \exists \ell \in \{1, \ldots, G\}$, such that $j \in \{\pi_\ell(1), \ldots, \pi_\ell(s_\ell)\}\}$, where $\pi_\ell$ is the same permutation as in Definition 4.1 for the set of indices $I_\ell$.*

*Proof.* The result follows from the mutual exclusiveness of $I_\ell$ in the constraints of equation 10 and Lemma 4.2. $\square$

## 5 EXPERIMENTS

This section addresses the following questions: 1) How can we efficiently update the vision encoder while preserving its generic knowledge? 2) Does updating the vision encoder separately and then reintegrating it into the corresponding VLM enhance downstream VQA performance? 3) How does updating the vision encoder separately compare to adapting the large language model in terms of VQA performance? 4) How does our method, LoRSU, compare to other parameter update methods in image classification and VQA tasks under different continual, few-shot, and offline settings?

## 5.1 SETTING

**Classification datasets. TSI**: We process the TSI Das et al. (2019) dataset as an image classification dataset where the target is to recognize the activity depicted in each image. We extract frames from videos and create a train set of approximately 10K images and a test set of approximately 5k images. We consider 22 represented classes of activities. **DALLE**: We consider the same 22 classes of activities represented in TSI and query DALL·E 2 to generate representative images of these activities. We extract 30 images per action totaling 660, all of them are designated for testing. **ImgNet**: We consider ImageNet (Deng et al., 2009) as a control set to measure how much CLIP models' performance deteriorates after being tuned on other datasets. **GTS** (Stallkamp et al., 2012) the German Traffic Sign dataset. Zhang et al. (2024) considered GTS as out of distribution for CLIP pretraining. **AIR** (Maji et al., 2013), a fine-grained aircraft classification dataset. For both GTS and AIR, CLIP zero shot performance is significantly lower than the performance of a linear classifier trained on ResNet50 features Radford et al. (2021). (**CAn**) (Wang et al., 2024) a recent dataset to examine the robustness of pretrained image encoders, it contains animal images with realistic spurious features such as unexpected backgrounds.

**Visual Question Answering datasets.** To evaluate how the examined VLM performs before and after the vision encoder update, we consider six visual question answering datasets: HM Kiela et al. (2020) hateful memes dataset designed to detect multimodal hateful memes. The rest of the datasets were created by converting multi-class classification datasets into VQA datasets with multiple choice responses and measuring the probability of the correct response; a common practice in VQA evaluation. The converted VQA datasets are based on images of **DALLE**, **TSI**, **GTS**, **AIR**, and **CAn**. The last four datasets (and their corresponding VQA versions) are used for fine-tuning either the visual encoder or the LLM.

**Training protocols. Offline**: We first consider an offline fine-tuning setting as a sanity check where the vision encoder is updated offline on the full training set of each of the six datasets. The goal is to asses the performance of CLIP before and after the update by different methods and the VLM responses when the updated vision backbone is plugged in. **Continual & few-shot**: We design this setting to imitate a realistic scenario where the model is updated on images where it makes mistakes with few-shot examples, and the process is to be repeated as long mistakes are shown. We follow the common practice in continual few-shot learning Panos et al. (2023) to construct the sequences. We divide the dataset into 5 sets of disjoint classes and consider 5 shot setting where only 5 training examples of each action is provided. Accuracy is measured on the full test set. In the Appendix we consider 50 shots and 20 shots settings. **Metrics**: We consider the zero shot accuracy of image classification and VQA as the benchmark baseline and we report the change in that accuracy on the test/control sets of the target dataset where adaptation is performed at the end of a training sequence; in that way, we measure the ability of the model to accumulate knowledge. We name this metric as **Target Improvement** accuracy. We also calculate the average change on all other test/control sets when updating on a specific dataset to estimate average forgetting of generic knowledge or possible positive backward transfer (De Lange et al., 2021); we call this metric as **Average Control Change** accuracy where 'control' refers to the control datasets we use to calculate the average accuracy change. Note that we do not consider any replay buffer (Chaudhry et al., 2019) of samples from classes of previous sessions as is common in previous works.

**Implementation details.** We refer to the Appendix B for implementation details.

**Models.** For our experiments, we consider the popular Vision Language Model LLaVA (1.5) (Liu et al., 2024b) that leverages a frozen CLIP image encoder. Specifically, LLaVA utilizes a frozen OpenAI-CLIP-L-14 Radford et al. (2021) with a LLM (Vicuna-7b (Chiang et al., 2023)). The two modules are connected through a two-layer MLP projector that aligns vision and text features. The LLM and the MLP projector are optimized during the visual instruction tuning process while CLIP remains frozen. LLaVA concatenates adjacent tokens from CLIP-L-14 and processes it with an MLP projector as input to LLama-2 (7B-chat) (Touvron et al., 2023); the MLP projector and the language model are optimized while the vision encoder remains frozen.

**Methods.** When fine-tuning CLIP, we fine-tune both visual and text encoders following (Goyal et al., 2023) with the same contrastive image language loss used in the pretraining of CLIP. We consider the following methods for fine-tuning. **F-FT**: Full fine-tuning of all model parameters. This can provide the best accuracy, but is prone to forgetting and overfitting. **F-EWC**: This is a variant of F-FT which is based on the popular Continual Learning method EWC (Kirkpatrick et al., 2017) where an

Table 3: **Target Improvement** (↑) and **Average Control Change** (↑) (in parentheses) based on **classification accuracy** for all baselines that fine-tune the CLIP-L-14. **Target-D** denotes the dataset that we fine-tune the visual encoder on.

| Target-D | Setting | FT Method | | | | | |
| --- | --- | --- | --- | --- | --- | --- | --- |
| | | LN | F-FT | F-EWC | LoRA | SPU | LoRSU |
| TSI | CL-5 | 17.7(−0.1) | 27.4(−1.8) | 18.6(−3.5) | 20.2(−4.8) | 19.6(−0.2) | 28.7(0.6) |
| | Offline | 61.2(−9.7) | 67.7(−2.6) | − | 65.3(−5.4) | 62.1(−2.0) | 66.7(−0.9) |
| GTS | CL-5 | 12.2(−0.2) | 11.1(−3.1) | 14.0(−2.8) | 8.7(−6.1) | 17.1(−0.4) | 21.7(0.1) |
| | Offline | 45.9(−49.9) | 46.4(−4.6) | − | 47.1(−15.9) | 46.5(−0.1) | 46.8(−0.2) |
| AIR | CL-5 | 4.9(−0.5) | 0.6(−0.4) | 3.8(−1.7) | 1.7(−0.4) | 6.6(0.1) | 7.7(0.2) |
| | Offline | 24.5(−4.4) | 41.7(−1.5) | − | 42.0(−2.0) | 32.2(−0.2) | 32.9(−0.1) |
| CAn | CL-5 | −1.3(−0.4) | −19.4(−5.2) | −14.6(−4.2) | −19.6(−11.2) | −3.3(−0.1) | −4.7(−0.6) |
| | Offline | 21.3(−2.4) | 27.2(−1.9) | − | 29.8(−4.4) | 21.8(0.1) | 29.3(−0.5) |
| Average | CL-5 | 8.4(−0.3) | 4.9(−2.6) | 5.5(−3.0) | 2.8(−5.6) | 10.0(−0.2) | **13.3**(**0.1**) |
| | Offline | 38.2(−16.6) | 45.8(−2.7) | − | **46.1**(−6.9) | 40.6(−0.5) | 43.9(−**0.5**) |

L2 regularization is added to penalize changes on parameters deemed important for previous tasks. **LN**: Optimization of the layer norm parameters of the transformer, an adaptation of Shysheya et al. (2022). This approach modifies a very small fraction of parameters and has been shown to be a robust approach for few-shot updates (Panos et al., 2023). **LoRA**: we consider Low rank updates (Hu et al., 2021) applied to all transformer layers. **SPU**: Selective Parameters Updates Zhang et al. (2024) is a recent method proposed to continually update CLIP with minimal forgetting and generic knowledge loss. **LoRSU:** This is our method described in section 4. We always report the zero-shot performance of the model (without training), we refer to this as **Zr-shot**. Finally, we add the suffixes **-V** or **-L** to a method's name for denoting the fine-tuning of the vision encoder or the LLM, respectively.

## 5.2 RESULTS

**Image Classification Results.** First we evaluate the image classification accuracy based on CLIP adapted backbones with the different methods, results reported in Table 3 in form of Target Improvement on the target dataset and Average Control Change on Imagenet and DALLE datasets. When offline updates are performed, full fine-tunning and LoRA succeeds in improving the performance on the target data set, however both F-Ft and LoRA incur considerable forgetting and deterioration on other datasets accuracy even when one session of offline updates is conducted. SPU and LoRSU succeed in improving the performance on the target dataset while having minimal forgetting on generic model knowledge. LoRSU on overage achieves best target dataset performance with minimal forgetting compared to all other method. For example on GTS, LoRSU improves the target accuracy by 46.8 compared to 47.1 by LoRA, however the performance on generic knowledge by LoRA dropped by 15.9 compared to a negligible forgetting of 0.2 by LoRSU. Layer norm updates (LN) achieves he least improvement on the target dataset compared to other methods except EWC. With EWC, the performance on both current and previous datasets, indicating difficulties on the optimization of a large model with such strong regularization penalties. Regarding the continual few-shot updates, we see that LoRSU achieves the best results on the target dataset by a margin of at least 3% to the second best method while not only having no forgetting at all but improves accuracy on the control datasets on average. *Our method LoRSU improves the classification accuracy on the target dataset with a minimal deterioration on other datasets performances (< 1%) on both offline and continual few shot updates, with no replay of any samples from previous or other tasks.* This is a results of localizing the updates to only a small set of parameters that are relevant to the data at hand, while keeping the rest of the parameters intact.

**VQA performance after offline and continual few-shot updates of CLIP.** After updating CLIP model on image classification tasks, we take the updated vision encoder and plug it back in LLaVA model, i.e. simply replace the frozen vision encoder of LLaVA with the one the we have **separately** updated. We evaluate the VQA performance on the target dataset as well as other datasets and report the Average Improvement (on the target dataset) and Average Control Change on other datasets in Table 3 for both continual few-shot and offline settings. The first observation is that successful improvements on the classification performance and reflected in VQA performance of LLaVA in spite

Table 4: **Target Improvement** (↑) and **Average Control Change** (↑) (in parentheses) based on **VQA accuracy** for all baselines that fine-tune the CLIP-L-14. **Target-D** denotes the dataset used for fine-tuning, F-EWC only applied to continual setting.

| Target-D | Setting | FT Method | | | | | |
|---|---|---|---|---|---|---|---|
| | | LN | F-FT | F-EWC | LoRA | SPU | LoRSU |
| TSI | CL-5 | 0.4(−0.6) | 7.8(−3.0) | 1.5(−4.6) | 2.3(−4.9) | −0.6(−0.3) | 5.1(0.2) |
| | Offline | 18.0(−4.0) | 25.7(−2.7) | − | 23.6(−3.6) | 17.9(−0.5) | 22.4(−0.8) |
| GTS | CL-5 | 2.8(−0.1) | 1.7(−1.9) | −0.2(−3.7) | 1.4(−3.3) | 4.7(0.2) | 5.9(0.1) |
| | Offline | 11.4(−9.3) | 14.9(−3.8) | − | 15.1(−3.5) | 14.9(−0.4) | 15.5(−0.4) |
| AIR | CL-5 | 0.1(−0.7) | 0.6(−2.1) | −0.1(−2.6) | 2.4(−2.0) | 3.6(0.1) | 4.4(−0.0) |
| | Offline | 0.0(−1.7) | 3.4(−0.8) | − | 7.4(−1.7) | 9.4(−0.3) | 10.0(−0.1) |
| CAn | CL-5 | −1.3(−0.8) | −7.0(−2.9) | −8.1(−5.8) | −10.2(−4.3) | −2.4(−0.2) | −1.7(−0.3) |
| | Offline | −0.2(−2.6) | 4.0(−1.8) | − | 1.4(−2.9) | 1.5(−0.2) | 2.3(−0.3) |
| Average | CL-5 | 0.5(−0.6) | 0.8(−2.5) | −1.7(−4.2) | −1.0(−3.6) | 1.3(−0.1) | **3.4(0.0)** |
| | Offline | 7.3(−4.4) | 12.0(−2.3) | − | 11.9(−2.9) | 10.9(−**0.4**) | **12.6**(−0.4) |

Table 5: **LLM vs. V. Encoder FT**: We compare performance between the fine-tuned vision encoder and the LLM. We report the **Target Improvement** (↑) and **Average Control Change** (↑) (in parentheses) based on **VQA accuracy**. 'V' and 'L' indicates whether a parameter-efficient fine-tuning method adapts the vision encoder or the LLM, respectively.'+' denotes the fine tuning of the MLP projector along with the fine-tuning of LLM.

| Target-D | Setting | FT Method | | | | |
|---|---|---|---|---|---|---|
| | | LoRA-L | LoRSU-L | LoRA-L+ | LoRSU-L+ | LoRSU-V |
| TSI | CL-5 | 5.3(0.5) | 1.2(0.5) | 15.4(−0.9) | 8.8(0.1) | 5.1(0.2) |
| | Offline | 14.4(0.1) | 8.5(0.5) | 20.2(−2.6) | 25.2(−4.4) | 22.4(−0.8) |
| GTS | CL-5 | −4.5(−0.5) | −0.3(0.1) | −2.6(−2.2) | 2.7(−0.3) | 5.9(0.1) |
| | Offline | −5.0(−0.8) | −1.3(−0.2) | −10.6(−4.8) | 0.0(−4.8) | 15.5(−0.4) |
| AIR | CL-5 | −1.1(−0.0) | −0.3(0.7) | 9.3(−0.5) | 13.7(−0.1) | 4.4(−0.0) |
| | Offline | 4.6(−0.5) | −0.5(0.0) | 11.6(−2.6) | 15.2(−0.3) | 10.0(−0.1) |
| CAn | CL-5 | −2.2(−0.6) | −0.4(−0.2) | 0.5(−1.1) | 0.9(−1.2) | −1.7(−0.3) |
| | Offline | −3.6(−0.6) | −1.2(0.2) | 1.9(−1.6) | 0.6(−1.0) | 2.3(−0.3) |
| Average | CL-5 | −0.6(−0.2) | 0.0(0.3) | 5.7(−1.2) | 6.5(−0.3) | 3.4(0.0) |
| | Offline | 12.3(−0.4) | 12.3(−0.4) | 12.3(−0.4) | 12.3(−0.4) | 12.6(−0.4) |

of the separate update of the vision encoder with CLIP contrastive loss. Similar trend of conclusions can be made on the relevant performance of different methods. F-FT, LoRA, SPU and LoRSU. In spite of being low rank update, LoRA incurs higher forgetting and deterioration on other tasks performance than full fine-tuning (F-FT). LoRSU and SPU achieve stable and local updates with minimal deterioration on other tasks ( $< 1\%$ ), notably LoRSU achieves higher improvements than SPU on the target dataset an evidence of the flexibility brought by allowing structural updates on different layers. With regard to few shot updates, the improvements on the target dataset is less with all methods deteriorating the performance on the target datasets for the CAn dataset, due to the very challenging nature of updates with only 5 examples. However, LoRSU is the least affected and the method with the largest magnitude of improvements on all target datasets. We can conclude that LoRSU applied to the vision encoder separately improves the VQA performance of LLaVA the most with a minimal deterioration on other tasks.

**How does updating the vision encoder separately compares to update the LLM on VQA tasks** Here we fine-tune the LLM model of LLaVA using the standard perplexity loss on the VQA datasets and consider updating both the LLM (-L) and the non-linear projection (-L+) on both offline and continual few-shot updates (with no replay). We compare LoRSU with LoRA which is the standard method for updating the LLM and it requires reasonable compute resources. Updating the LLM alone without updating the projection layer leads to different results on different datasets, for some datasets it results in performance improvements while on other results on significant performance

deterioration. Updating the LLM and the projection layer (denoted by L+), results on significant improvements on TSI and AIR datasets for both LoRA-L and LoRSU-L. LoRSU-L+ achieves the best performance on the target dataset with comparable slight deterioration on the other datasets. *This indicates that LoRSU is not specific to the vision encoder and can be applied to any transformer model, either LLM or VIT.* Now, to answer the main question, LoRSU applied to the vision encoder a steady and significant improvements on all datasets for both offline and continual few-shot updates. On CAn few-shot updates, we do not see any improvement or deterioration on the VQA performance; we assume this might be due to the in-distribution nature of the CAn animal images and their textual information. Notably, updating the vision encoder separately incurs the least deterioration on VQA performance of other tasks on both offline and continual few-shot updates. We can state that LoRSU is an effective parameter efficient tuning method for both the LLM and the vision encoder, and that stable updates with minimal performance deterioration on generic knowledge can be achieved when LoRSU is applied to the vision encoder separately.

**What is the computation advantage of updating the vision encoder separately with LoRSU?** Figure 2 reports the compute cost in terms of TFlops (teraflops per second) incurred by updating the LLM and LLM with the MLP projector compared to updating the vision encoder separately with LoRSU.

The figure also reports the average time needed for one epoch of training in minutes, the percentage of updated parameters, and the VQA accuracy on GTS dataset. LoRSU and LoRA on the LLM and projector require comparable computation cost. LoRSU-v needs 0.36 TFlops compared to 9.0 TFlops by LoRA-L+ and 9.1 TFlops by LoRSU-L+. LoRSU-v takes only **5.5** minutes per epoch compared to 76.4 and 78.6 by LoRA-L+ and LoRSU-L+ respectively. *LoRSU is an efficient and effective method, and when applied to the vision encoder separately it achieves a significant and stable performance improvement on VQA tasks with less than 10x compute and training time! compared to updating the LLM for VQA tasks.*

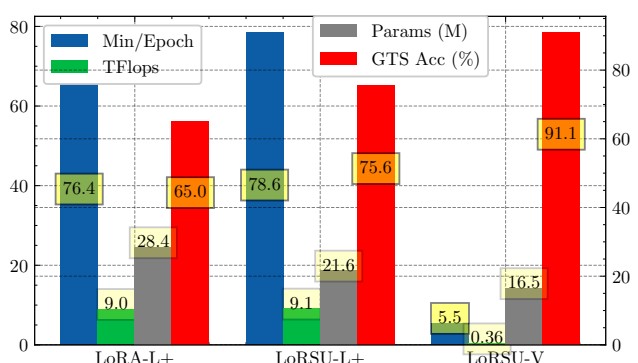

Figure 2: TFlops, training time (average minutes per epoch ) and performance comparison between the fine-tuned LLM and the fine-tuned visual encoder using our method LoRSU (LoRSU-V) for the offline setting on GTS dataset. We report results based on a single NVidia A100 GPU.

## 6  CONCLUSION

In this work, we investigated the limitations of CLIP on out-of-distribution and fine-grained benchmarks and noted how these weaknesses are inherited by the VLMs that utilize CLIP's embeddings. To address this, we propose a novel approach: updating the vision encoder separately, specifically on data where CLIP fails. Remarkably, this strategy significantly corrects VLM mistakes on previously unseen images from the same data. We further introduce a parameter-efficient tuning method, LoRSU, that not only targets efficiency but also ensures the preservation of the model's generic knowledge through localized and structured updates. Our method, LoRSU, can be successfully applied to both the LLM and the vision encoder. In our experiments, LoRSU is the only method to systematically improve the classification performance of CLIP as well as the VLM performance on VQA tasks, with the least deterioration in performance on other tasks, even in the challenging but realistic continual few-shot setting with no replay of previous tasks' data. Our approach hence strikes a strong balance between efficiency, effectiveness, and robustness, achieving new state-of-the-art results. Due to limitations in compute, we focus on CLIP and LLaVA. We plan to scale our work to other VLMs and vision encoders, as we believe our conclusion scales well since our method is generic to any transformer model.

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

# A APPENDIX

## A.1 ADDITIONAL RESULTS

**Deatailed accuracues** we report here the detailed results of other experimental setting such as classification and VQA accuracies for 5/20/50 shots and offline settings. Classification accuracies for each of the four datasets used for fine-tuning can be found in Tables 6, 7, 8, and 9. The corresponding VQA accuracies can be found in Tables 10, 11, 12, and 13. VQA accuracies for the comparison between fine-tuned LLM and fine-tuned visual encoder are in Tables 14, 15, 16, and 17. These accuracies are used to build Tables 3, 4, and 5 in the main paper.

**Rank ablation** We investigate how the choice of rank affects the classification accuracy in Tables 21 and 22.

## A.2 PROMPTS USED TO GENERATE IMAGES FROM DALL·E 2

We generated images from DALL·E 2 using OpenAI python package and we used the prompt "*A person {a}*" where $a \in \{$ *using a white coffee machine, eating, cutting bread, stirring the pot, holding a glass, watching TV, holding a bottle, walking, making tea, cutting food, holding a cup, using a laptop, lying down, holding a can, person holding a black kettle, reading a book, cleaning up, sitting down, using a tablet, boiling water in a black kettle, using a cordless phone, washing dishes*$\}$.

Table 6: Classification accuracy for CLIP-L-14 model for various fine-tuning methods. The model is fine-tuned on each session (5 in total) of **TSI** dataset using **5/10/50 shots** per session. Finally, we consider the **offline** setting too.

| | | | FT Method | | | | | |
|---|---|---|---|---|---|---|---|---|
| **Setting** | **Dataset** | **Zr-Shot** | **LN** | **F-FT** | **F-EWC** | **LoRA** | **SPU** | **LoRSU** |
| **CL-5 shots** | **ImgNet** | 76.6 | 76.3 | 73.9 | 72.1 | 69.7 | 76.7 | 76.7 |
| | **DALLE** | 90.9 | 90.9 | 90.0 | 88.5 | 88.3 | 90.3 | 92.1 |
| | **TSI** | 13.2 | 30.9 | 40.6 | 31.8 | 33.4 | 32.8 | 41.9 |
| **CL-20 shots** | **ImgNet** | 76.6 | 75.0 | 74.2 | 75.8 | 72.1 | 76.5 | 76.5 |
| | **DALLE** | 90.9 | 92.1 | 89.1 | 89.4 | 88.2 | 91.1 | 92.3 |
| | **TSI** | 13.2 | 42.8 | 56.3 | 49.1 | 52.6 | 47.6 | 49.7 |
| **CL-50 shots** | **ImgNet** | 76.6 | 72.2 | 73.0 | 74.3 | 69.3 | 76.4 | 76.4 |
| | **DALLE** | 90.9 | 87.3 | 88.5 | 89.4 | 77.3 | 91.7 | 92.3 |
| | **TSI** | 13.2 | 43.2 | 31.2 | 54.0 | 54.5 | 52.8 | 58.0 |
| **Offline** | **ImgNet** | 76.6 | 65.8 | 73.9 | — | 72.2 | 76.3 | 74.6 |
| | **DALLE** | 90.9 | 82.3 | 88.3 | — | 84.4 | 87.3 | 91.1 |
| | **TSI** | 13.2 | 74.4 | 80.9 | — | 78.5 | 75.3 | 79.9 |

| | Test Dataset (Top-1 Acc %) | | | | |
|---|---|---|---|---|---|
| **Pretraining Dataset** | **Blur_bg** | **Blur_obj** | **Color** | **Rand_bg** | **Seg_img** |
| ViT-B-16 ( ImageNet) | 88.4 | 90.8 | 66.5 | 17.2 | 49.0 |
| ViT-B-16 (XImageNet-12) | 71.51 | 70.21 | 74.14 | 38.01 | 78.7 |
| CLIP-ViT-B-16 (DATACOMP) | 98.9 | 97.5 | 98.6 | 42.4 | 95.4 |
| CLIP-ViT-L-14 (OpenAI) | 98.9 | 98.2 | 98.3 | 52.5 | 95.7 |

Table 18: Performance on XImageNet-12 benchmark with ViT-B and ViT-L considering different pretraining settings. CLIP pretraining with DATACOMP is quite robust to various shifts.

# B  IMPLEMENTATION DETAILS

- We use a single A100 GPU for the experiments.
- We use Adam Kingma (2014) as an optimizer for the fine tuning of CLIP-L-14 and AdamW Loshchilov (2017) for the fine-tuning of LLaVA's LLM. We also use a learning rate scheduler of Cosine Annealing with Warmup for all methods.
- We use batch size 8 for the few shot experiments and batch size 64 for the offline ones.
- We run all experiments using 10 epochs.
- For Lora, we use rank $r = 64$ for all experiments.
- For SPU, we use sparsity=15% for all experiments.
- For LoRSU we use sparsity=10%, rank=64, and we pick the top-2 attention heads for all experiments.
- For LoRSU and SPU, the binary mask for the first MLP layer is constructed by using either 800 data points to compute gradients in the offline setting, or all available data points from the current task's dataset in the CL-few shot setting.
- For all VQA datasets, we measure performance based on accuracy of the predicted answers of LLaVA.
- We converted DALLE, TSI, GTS, AIR, and CAn as a multiple choice VQA problem where each question has five choices and the VLM is asked to choose the right one.

Table 7: Classification accuracy for CLIP-L-14 model for various fine-tuning methods. The model is fine-tuned on each session (5 in total) of **GTS** dataset using **5/10/50 shots** per session. Finally, we consider the **offline** setting too.

| Setting | Dataset | Zr-Shot | FT Method | | | | | |
|---|---|---|---|---|---|---|---|---|
| | | | LN | F-FT | F-EWC | LoRA | SPU | LoRSU |
| **CL-5 shots** | **ImgNet** | 76.6 | 76.2 | 71.6 | 73.1 | 68.3 | 76.6 | 76.7 |
| | **DALLE** | 90.9 | 90.9 | 89.8 | 88.9 | 87.1 | 90.2 | 91.1 |
| | **GTS** | 52.4 | 64.6 | 63.5 | 66.4 | 61.1 | 69.5 | 74.1 |
| **CL-20 shots** | **ImgNet** | 76.6 | 74.6 | 64.8 | 68.5 | 60.1 | 76.6 | 76.6 |
| | **DALLE** | 90.9 | 92.1 | 87.7 | 88.1 | 83.8 | 91.2 | 91.5 |
| | **GTS** | 52.4 | 71.5 | 59.9 | 68.8 | 67.3 | 80.4 | 74.7 |
| **CL-50 shots** | **ImgNet** | 76.6 | 69.1 | 71.1 | 62.6 | 58.7 | 76.5 | 76.5 |
| | **DALLE** | 90.9 | 88.5 | 89.4 | 87.9 | 82.6 | 91.2 | 91.4 |
| | **GTS** | 52.4 | 74.7 | 66.4 | 68.9 | 66.2 | 81.6 | 83.2 |
| **Offline** | **ImgNet** | 76.6 | 23.9 | 70.0 | – | 54.8 | 75.9 | 75.6 |
| | **DALLE** | 90.9 | 43.9 | 88.2 | – | 80.9 | 91.4 | 91.5 |
| | **GTS** | 52.4 | 98.3 | 98.8 | – | 99.5 | 98.9 | 99.2 |

Table 8: Classofication accuracy for CLIP-L-14 model for various fine-tuning methods. The model is fine-tuned on each session (5 in total) of **AIR** dataset using **5/10/50 shots** per session. Finally, we consider the **offline** setting too.

| Setting | Dataset | Zr-Shot | FT Method | | | | | |
|---|---|---|---|---|---|---|---|---|
| | | | LN | F-FT | F-EWC | LoRA | SPU | LoRSU |
| **CL-5 shots** | **ImgNet** | 76.6 | 76.0 | 75.5 | 74.8 | 74.9 | 76.8 | 76.8 |
| | **DALLE** | 90.9 | 90.5 | 91.2 | 89.3 | 91.7 | 90.9 | 91.1 |
| | **AIR** | 33.4 | 38.3 | 34.0 | 37.2 | 35.1 | 40.0 | 41.1 |
| **CL-20 shots** | **ImgNet** | 76.6 | 73.9 | 75.8 | 72.7 | 70.8 | 76.9 | 76.6 |
| | **DALLE** | 90.9 | 90.2 | 90.8 | 89.0 | 88.3 | 90.6 | 92.0 |
| | **AIR** | 33.4 | 38.5 | 36.5 | 38.5 | 35.6 | 41.8 | 43.0 |
| **CL-50 shots** | **ImgNet** | 76.6 | 70.3 | 74.9 | 74.2 | 65.6 | 76.4 | 76.3 |
| | **DALLE** | 90.9 | 88.9 | 89.7 | 89.1 | 88.3 | 89.5 | 90.6 |
| | **AIR** | 33.4 | 40.4 | 39.2 | 40.6 | 36.5 | 43.3 | 44.2 |
| **Offline** | **ImgNet** | 76.6 | 70.2 | 75.5 | – | 74.5 | 76.2 | 76.0 |
| | **DALLE** | 90.9 | 88.5 | 88.9 | – | 88.9 | 90.8 | 91.2 |
| | **AIR** | 33.4 | 57.9 | 75.1 | – | 75.4 | 65.6 | 66.3 |

Table 9: Classification accuracy for CLIP-L-14 model for various fine-tuning methods. The model is fine-tuned on each session (5 in total) of **CAn** dataset using extbf5/10/50 shots per session. Finally, we consider the **offline** setting too.

| Setting | Dataset | Zr-Shot | FT Method | | | | | |
| | | | LN | F-FT | F-EWC | LoRA | SPU | LoRSU |
|---|---|---|---|---|---|---|---|---|
| **CL-5 shots** | **ImgNet** | 76.6 | 75.8 | 67.3 | 69.7 | 60.1 | 76.3 | 75.9 |
| | **DALLE** | 90.9 | 90.9 | 89.8 | 89.4 | 85.0 | 90.9 | 90.3 |
| | **CAn** | 64.1 | 62.8 | 44.7 | 49.5 | 44.5 | 60.8 | 59.4 |
| **CL-20 shots** | **ImgNet** | 76.6 | 73.0 | 62.8 | 67.2 | 58.6 | 75.7 | 73.8 |
| | **DALLE** | 90.9 | 90.5 | 86.8 | 85.8 | 81.7 | 90.8 | 89.2 |
| | **CAn** | 64.1 | 59.6 | 54.6 | 56.7 | 46.3 | 60.2 | 59.7 |
| **CL-50 shots** | **ImgNet** | 76.6 | 69.1 | 57.8 | 59.6 | 57.2 | 74.6 | 70.2 |
| | **DALLE** | 90.9 | 88.5 | 83.5 | 69.1 | 86.1 | 90.5 | 88.8 |
| | **CAn** | 64.1 | 62.1 | 55.1 | 58.0 | 53.8 | 65.4 | 58.9 |
| **Offline** | **ImgNet** | 76.6 | 71.5 | 73.2 | — | 68.7 | 76.0 | 75.5 |
| | **DALLE** | 90.9 | 91.2 | 90.6 | — | 90.0 | 91.8 | 90.9 |
| | **CAn** | 64.1 | 85.4 | 91.3 | — | 93.9 | 85.9 | 93.4 |

Table 10: VQA accuracy scores (%) for LLaVA with the pretrained or fine-tuned CLIP CLIP-L-14. All baselines use **TSI** dataset for fine-tuning (the LLM remains frozen).

| Setting | FT Method | VQA Datasets (Acc %) | | | | | |
| | | HM | DALLE | TSI | GTS | AIR | CAn |
|---|---|---|---|---|---|---|---|
| | **Zr-Shot** | 61.2 | 91.1 | 53.1 | 75.6 | 60.4 | 82.7 |
| **CL-5 shots** | **LN** | 61.9 | 90.3 | 53.5 | 74.6 | 59.5 | 81.8 |
| | **F-FT** | 60.4 | 89.4 | 60.9 | 71.5 | 55.1 | 79.5 |
| | **F-EWC** | 59.9 | 87.3 | 54.6 | 69.0 | 56.2 | 75.6 |
| | **LoRA** | 61.1 | 88.0 | 55.4 | 70.8 | 53.3 | 73.3 |
| | **SPU** | 61.0 | 90.9 | 52.5 | 75.1 | 60.3 | 82.2 |
| | **LoRSU** | 61.5 | 91.4 | 58.2 | 76.1 | 60.5 | 82.5 |
| **CL-20 shots** | **LN** | 61.7 | 90.8 | 58.8 | 73.7 | 57.7 | 79.6 |
| | **F-FT** | 61.3 | 89.7 | 68.6 | 73.1 | 57.1 | 79.7 |
| | **F-EWC** | 59.3 | 79.1 | 58.7 | 63.1 | 39.1 | 70.5 |
| | **LoRA** | 60.6 | 89.1 | 66.7 | 69.4 | 53.1 | 76.1 |
| | **SPU** | 62.1 | 90.5 | 62.0 | 75.4 | 59.7 | 82.2 |
| | **LoRSU** | 61.5 | 90.9 | 65.7 | 75.3 | 60.3 | 82.3 |
| **CL-50 shots** | **LN** | 61.1 | 87.4 | 58.3 | 71.5 | 53.8 | 78.5 |
| | **F-FT** | 61.6 | 89.2 | 70.0 | 40.5 | 33.4 | 45.7 |
| | **F-EWC** | 58.5 | 66.1 | 70.9 | 41.5 | 35.4 | 47.7 |
| | **LoRA** | 60.8 | 89.5 | 71.5 | 66.4 | 52.7 | 76.4 |
| | **SPU** | 61.8 | 90.5 | 65.6 | 75.1 | 59.0 | 82.3 |
| | **LoRSU** | 61.9 | 90.6 | 70.8 | 75.5 | 59.3 | 82.2 |
| **Offline** | **LN** | 61.2 | 87.4 | 71.1 | 70.6 | 53.1 | 78.6 |
| | **F-FT** | 61.5 | 89.1 | 78.8 | 71.9 | 55.4 | 79.6 |
| | **LoRA** | 60.6 | 87.9 | 76.7 | 69.6 | 57.6 | 77.2 |
| | **SPU** | 62.2 | 91.5 | 71.0 | 75.2 | 57.9 | 81.5 |
| | **LoRSU** | 62.1 | 91.1 | 75.5 | 74.7 | 58.0 | 81.2 |

Table 11: VQA accuracy scores (%) for LLaVA with the pretrained or fine-tuned CLIP CLIP-L-14. All baselines use **GTS** dataset for fine-tuning (the LLM remains frozen).

| Setting | FT Method | VQA Datasets (Acc %) | | | | | |
|---|---|---|---|---|---|---|---|
| | | HM | DALLE | TSI | GTS | AIR | CAn |
| | **Zr-Shot** | 61.2 | 91.1 | 53.1 | 75.6 | 60.4 | 82.7 |
| **CL-5 shots** | **LN** | 62.0 | 91.5 | 53.1 | 78.4 | 60.1 | 81.1 |
| | **F-FT** | 61.6 | 88.2 | 54.5 | 77.3 | 57.9 | 76.8 |
| | **F-EWC** | 61.1 | 88.8 | 55.5 | 75.4 | 55.8 | 68.9 |
| | **LoRA** | 62.0 | 86.8 | 54.7 | 77.0 | 55.8 | 72.7 |
| | **SPU** | 61.8 | 91.8 | 53.3 | 80.3 | 60.3 | 82.2 |
| | **LoRSU** | 61.9 | 91.5 | 52.9 | 81.5 | 60.5 | 82.1 |
| **CL-20 shots** | **LN** | 62.2 | 89.2 | 51.2 | 79.7 | 60.5 | 78.8 |
| | **F-FT** | 61.6 | 87.0 | 50.4 | 77.5 | 55.8 | 72.7 |
| | **F-EWC** | 60.5 | 81.4 | 49.2 | 78.0 | 54.8 | 40.0 |
| | **LoRA** | 61.7 | 83.0 | 55.4 | 81.7 | 50.0 | 68.2 |
| | **SPU** | 61.1 | 91.1 | 53.1 | 81.1 | 60.2 | 81.5 |
| | **LoRSU** | 61.2 | 91.8 | 52.9 | 83.9 | 60.2 | 81.7 |
| **CL-50 shots** | **LN** | 62.2 | 86.4 | 51.8 | 79.7 | 58.7 | 73.8 |
| | **F-FT** | 61.5 | 88.0 | 45.6 | 81.3 | 48.8 | 61.6 |
| | **F-EWC** | 53.4 | 39.1 | 18.1 | 61.5 | 21.8 | 22.6 |
| | **LoRA** | 61.8 | 87.6 | 47.6 | 75.2 | 48.8 | 61.6 |
| | **SPU** | 61.7 | 90.9 | 45.6 | 82.4 | 57.8 | 79.6 |
| | **LoRSU** | 61.2 | 90.6 | 52.9 | 84.2 | 60.4 | 81.4 |
| **Offline** | **LN** | 61.8 | 80.2 | 48.6 | 87.0 | 52.9 | 58.4 |
| | **F-FT** | 60.6 | 87.4 | 51.3 | 90.5 | 56.3 | 73.8 |
| | **LoRA** | 61.8 | 88.5 | 53.2 | 90.7 | 54.3 | 73.3 |
| | **SPU** | 61.4 | 90.9 | 53.7 | 90.5 | 59.8 | 80.8 |
| | **LoRSU** | 62.0 | 91.1 | 53.3 | 91.1 | 59.5 | 80.7 |

Table 12: VQA accuracy scores (%) for LLaVA with the pretrained or fine-tuned CLIP CLIP-L-14. All baselines use **AIR** dataset for fine-tuning (the LLM remains frozen).

| Setting | FT Method | VQA Datasets (Acc %) | | | | | |
|---|---|---|---|---|---|---|---|
| | | HM | DALLE | TSI | GTS | AIR | CAn |
| | **Zr-Shot** | 61.2 | 91.1 | 53.1 | 75.6 | 60.4 | 82.7 |
| **CL-5 shots** | **LN** | 61.5 | 92.4 | 51.8 | 73.1 | 60.5 | 81.2 |
| | **F-FT** | 60.9 | 89.8 | 49.7 | 73.1 | 61.0 | 79.9 |
| | **F-EWC** | 62.4 | 89.4 | 49.1 | 74.0 | 60.3 | 75.8 |
| | **LoRA** | 62.0 | 91.2 | 52.2 | 70.8 | 62.8 | 77.6 |
| | **SPU** | 61.3 | 92.0 | 52.9 | 75.4 | 64.0 | 82.6 |
| | **LoRSU** | 61.2 | 91.2 | 53.1 | 75.4 | 64.8 | 82.6 |
| **CL-20 shots** | **LN** | 61.9 | 89.7 | 50.4 | 70.1 | 60.3 | 77.0 |
| | **F-FT** | 61.0 | 90.9 | 52.3 | 70.5 | 68.5 | 79.0 |
| | **F-EWC** | 58.6 | 82.1 | 48.9 | 69.7 | 59.9 | 76.9 |
| | **LoRA** | 62.0 | 90.3 | 52.4 | 67.1 | 57.6 | 74.5 |
| | **SPU** | 61.9 | 91.4 | 52.2 | 75.0 | 64.8 | 82.1 |
| | **LoRSU** | 61.6 | 91.4 | 53.4 | 75.0 | 69.8 | 82.3 |
| **CL-50 shots** | **LN** | 61.6 | 88.5 | 54.8 | 67.3 | 62.4 | 74.8 |
| | **F-FT** | 60.8 | 91.2 | 51.5 | 71.8 | 68.6 | 71.6 |
| | **F-EWC** | 56.9 | 74.2 | 48.5 | 69.8 | 42.9 | 51.6 |
| | **LoRA** | 62.1 | 87.9 | 50.6 | 64.4 | 61.6 | 74.4 |
| | **SPU** | 62.0 | 91.1 | 52.0 | 75.2 | 68.1 | 81.6 |
| | **LoRSU** | 61.4 | 90.8 | 52.5 | 74.5 | 69.5 | 81.5 |
| **Offline** | **LN** | 62.8 | 90.5 | 54.9 | 70.2 | 60.4 | 76.6 |
| | **F-FT** | 62.0 | 90.9 | 53.7 | 73.3 | 63.8 | 80.0 |
| | **LoRA** | 61.5 | 90.2 | 54.0 | 71.0 | 67.8 | 78.3 |
| | **SPU** | 61.9 | 91.5 | 52.5 | 75.1 | 69.8 | 81.3 |
| | **LoRSU** | 62.4 | 91.4 | 52.6 | 75.0 | 70.4 | 81.6 |

Table 13: VQA accuracy scores (%) for LLaVA with the pretrained or fine-tuned CLIP CLIP-L-14. All baselines use **CAn** dataset for fine-tuning (the LLM remains frozen).

| Setting | FT Method | VQA Datasets (Acc %) | | | | | |
| | | HM | DALLE | TSI | GTS | AIR | CAn |
|---|---|---|---|---|---|---|---|
| | **Zr-Shot** | 61.2 | 91.1 | 53.1 | 75.6 | 60.4 | 82.7 |
| **CL-5 shots** | **LN** | 61.5 | 90.5 | 52.1 | 74.7 | 58.7 | 81.4 |
| | **F-FT** | 61.2 | 89.7 | 50.5 | 71.1 | 54.6 | 75.7 |
| | **F-EWC** | 60.6 | 86.2 | 49.3 | 69.4 | 46.7 | 74.6 |
| | **LoRA** | 61.0 | 91.1 | 47.6 | 67.4 | 52.6 | 72.5 |
| | **SPU** | 61.6 | 90.9 | 53.1 | 74.4 | 60.2 | 80.3 |
| | **LoRSU** | 61.7 | 91.2 | 52.5 | 74.7 | 59.6 | 81.0 |
| **CL-20 shots** | **LN** | 60.8 | 90.0 | 53.7 | 73.4 | 58.9 | 80.3 |
| | **F-FT** | 61.2 | 90.6 | 46.0 | 70.3 | 55.3 | 74.9 |
| | **F-EWC** | 59.6 | 89.2 | 48.6 | 71.3 | 55.7 | 76.1 |
| | **LoRA** | 61.5 | 89.5 | 47.8 | 70.0 | 52.8 | 79.7 |
| | **SPU** | 61.6 | 91.2 | 52.9 | 75.2 | 58.4 | 81.6 |
| | **LoRSU** | 62.1 | 91.7 | 52.2 | 75.1 | 58.0 | 82.0 |
| **CL-50 shots** | **LN** | 61.9 | 88.3 | 50.0 | 71.3 | 57.4 | 80.2 |
| | **F-FT** | 60.5 | 90.0 | 48.0 | 71.3 | 55.5 | 79.3 |
| | **F-EWC** | 60.0 | 45.9 | 49.9 | 73.5 | 51.2 | 75.2 |
| | **LoRA** | 60.8 | 90.5 | 48.3 | 65.8 | 54.8 | 82.3 |
| | **SPU** | 61.6 | 90.9 | 51.8 | 73.3 | 58.8 | 83.8 |
| | **LoRSU** | 61.8 | 91.7 | 51.9 | 74.5 | 57.1 | 82.7 |
| **Offline** | **LN** | 60.9 | 89.2 | 50.6 | 71.8 | 55.9 | 82.5 |
| | **F-FT** | 62.1 | 91.5 | 49.9 | 71.5 | 57.6 | 86.7 |
| | **LoRA** | 62.0 | 90.3 | 48.6 | 69.8 | 56.1 | 84.1 |
| | **SPU** | 61.8 | 91.2 | 52.7 | 75.0 | 59.5 | 84.2 |
| | **LoRSU** | 61.8 | 91.1 | 52.6 | 75.1 | 59.4 | 85.0 |

Table 14: Accuracy scores (%) for LLaVA. We fine-tune the LLM using LoRSU, and LoRA on the **TSI** dataset under different settings (the visual encoder remains frozen) and we compare its performance to our method LoRSU that fine-tunes the visual encoder (LoRSU-V). The suffix 'L' indicates that the method fine-tunes the LLM and 'L+' that the method fine tunes both the MLP projector and LLM.

| Setting | PEFT Method | VQA Datasets (Acc %) | | | | | |
|---|---|---|---|---|---|---|---|
| | | HM | DALLE | TSI | GTS | AIR | CAn |
| | **Zr-Shot** | 61.2 | 91.1 | 53.1 | 75.6 | 60.4 | 82.7 |
| **CL-5 shots** | **LoRA-L** | 63.3 | 91.2 | 58.4 | 75.7 | 60.7 | 82.7 |
| | **LoRSU-L** | 63.5 | 91.1 | 54.3 | 75.7 | 60.2 | 83.0 |
| | **LoRA-L+** | 59.9 | 90.9 | 68.5 | 75.9 | 58.3 | 81.6 |
| | **LoRSU-L+** | 63.0 | 89.7 | 61.9 | 76.0 | 59.8 | 83.2 |
| | **LoRSU-V** | 61.4 | 91.2 | 57.7 | 75.6 | 60.1 | 82.2 |
| **CL-20 shots** | **LoRA-L** | 64.2 | 91.4 | 60.5 | 75.9 | 60.0 | 82.2 |
| | **LoRSU-L** | 65.0 | 91.7 | 56.3 | 75.2 | 60.2 | 83.0 |
| | **LoRA-L+** | 63.2 | 86.4 | 65.3 | 75.9 | 59.8 | 81.2 |
| | **LoRSU-L+** | 63.7 | 86.2 | 69.8 | 76.5 | 51.4 | 78.1 |
| | **LoRSU-V** | 61.6 | 90.5 | 63.7 | 75.3 | 59.6 | 82.3 |
| **CL-50 shots** | **LoRA-L** | 63.0 | 90.8 | 64.5 | 75.9 | 59.8 | 82.2 |
| | **LoRSU-L** | 64.0 | 91.7 | 58.6 | 75.7 | 60.5 | 83.3 |
| | **LoRA-L+** | 60.0 | 80.0 | 63.8 | 74.7 | 58.0 | 78.0 |
| | **LoRSU-L+** | 64.8 | 83.3 | 63.8 | 76.4 | 58.4 | 82.6 |
| | **LoRSU-V** | 61.9 | 90.6 | 69.3 | 75.5 | 59.0 | 81.7 |
| **Offline** | **LoRA-L** | 62.9 | 91.5 | 67.5 | 76.1 | 58.8 | 82.0 |
| | **LoRSU-L** | 63.2 | 91.8 | 61.6 | 75.4 | 60.4 | 82.7 |
| | **LoRA-L+** | 60.5 | 88.5 | 73.3 | 75.8 | 53.4 | 79.6 |
| | **LoRSU-L+** | 52.6 | 90.2 | 78.3 | 76.3 | 51.0 | 78.7 |
| | **LoRSU-V** | 62.1 | 91.1 | 75.5 | 74.7 | 58.0 | 81.2 |

Table 15: Accuracy scores (%) for LLaVA. We fine-tune the LLM using LoRSU, and LoRA on the **GTS** dataset under different settings (the visual encoder remains frozen) and we compare its performance to our method LoRSU that fine-tunes the visual encoder (LoRSU-V). The suffix 'L' indicates that the method fine-tunes the LLM and 'L+' that the method fine tunes both the MLP projector and LLM.

| Setting | PEFT Method | VQA Datasets (Acc %) | | | | | |
|---|---|---|---|---|---|---|---|
| | | HM | DALLE | TSI | GTS | AIR | CAn |
| | **Zr-Shot** | 61.2 | 91.1 | 53.1 | 75.6 | 60.4 | 82.7 |
| **CL-5 shots** | **LoRA-L** | 61.2 | 91.4 | 52.0 | 71.1 | 59.7 | 81.5 |
| | **LoRSU-L** | 61.8 | 91.1 | 53.6 | 75.3 | 60.2 | 82.3 |
| | **LoRA-L+** | 56.0 | 90.6 | 50.2 | 73.0 | 60.1 | 80.8 |
| | **LoRSU-L+** | 62.5 | 90.9 | 52.4 | 78.3 | 60.1 | 81.1 |
| | **LoRSU-V** | 61.8 | 91.5 | 53.2 | 80.0 | 60.5 | 82.2 |
| **CL-20 shots** | **LoRA-L** | 62.1 | 92.0 | 52.2 | 70.9 | 59.5 | 82.3 |
| | **LoRSU-L** | 61.1 | 91.1 | 53.3 | 75.1 | 60.2 | 82.6 |
| | **LoRA-L+** | 55.4 | 91.4 | 50.0 | 69.1 | 58.9 | 77.2 |
| | **LoRSU-L+** | 61.8 | 91.1 | 52.2 | 76.7 | 59.2 | 79.6 |
| | **LoRSU-V** | 61.7 | 90.9 | 52.8 | 83.7 | 60.3 | 81.7 |
| **CL-50 shots** | **LoRA-L** | 64.2 | 91.4 | 52.7 | 67.3 | 59.6 | 80.8 |
| | **LoRSU-L** | 60.5 | 91.2 | 52.6 | 74.8 | 60.2 | 82.1 |
| | **LoRA-L+** | 54.0 | 91.4 | 47.1 | 63.2 | 58.5 | 72.2 |
| | **LoRSU-L+** | 52.1 | 90.0 | 51.8 | 73.5 | 58.3 | 80.7 |
| | **LoRSU-V** | 61.5 | 90.5 | 53.0 | 85.3 | 60.7 | 81.8 |
| **Offline** | **LoRA-L** | 59.2 | 91.5 | 54.8 | 70.6 | 58.3 | 80.5 |
| | **LoRSU-L** | 62.0 | 90.9 | 52.5 | 74.3 | 60.2 | 82.1 |
| | **LoRA-L+** | 58.0 | 92.1 | 45.5 | 75.0 | 58.3 | 70.7 |
| | **LoRSU-L+** | 44.6 | 89.2 | 53.4 | 75.6 | 58.2 | 78.9 |
| | **LoRSU-V** | 62.0 | 91.1 | 53.3 | 91.1 | 59.5 | 80.7 |

Table 16: Accuracy scores (%) for LLaVA. We fine-tune the LLM using LoRSU, and LoRA on the **AIR** dataset under different settings (the visual encoder remains frozen) and we compare its performance to our method LoRSU that fine-tunes the visual encoder (LoRSU-V). The suffix 'L' indicates that the method fine-tunes the LLM and 'L+' that the method fine tunes both the MLP projector and LLM.

| Setting | PEFT Method | VQA Datasets (Acc %) | | | | | |
|---|---|---|---|---|---|---|---|
| | | HM | DALLE | TSI | GTS | AIR | CAn |
| | **Zr-Shot** | 61.2 | 91.1 | 53.1 | 75.6 | 60.4 | 82.7 |
| **CL-5 shots** | **LoRA-L** | 60.9 | 91.7 | 53.9 | 75.5 | 59.3 | 81.5 |
| | **LoRSU-L** | 63.3 | 91.5 | 55.0 | 75.0 | 60.1 | 82.6 |
| | **LoRA-L+** | 60.5 | 90.9 | 54.3 | 75.0 | 69.7 | 80.6 |
| | **LoRSU-L+** | 62.6 | 90.9 | 54.8 | 74.4 | 74.1 | 80.7 |
| | **LoRSU** | 61.6 | 91.8 | 52.8 | 75.8 | 64.1 | 82.9 |
| **CL-20 shots** | **LoRA-L** | 60.0 | 92.1 | 54.6 | 74.8 | 66.2 | 81.3 |
| | **LoRSU-L** | 63.3 | 91.8 | 54.1 | 74.2 | 60.5 | 81.6 |
| | **LoRA-L+** | 59.0 | 92.1 | 52.8 | 72.6 | 73.2 | 78.1 |
| | **LoRSU-L+** | 63.1 | 90.9 | 53.6 | 74.3 | 76.7 | 83.0 |
| | **LoRSU-V** | 62.0 | 91.1 | 52.3 | 75.1 | 67.4 | 82.0 |
| **CL-50 shots** | **LoRA-L** | 59.2 | 92.1 | 55.3 | 73.9 | 60.4 | 82.1 |
| | **LoRSU-L** | 62.9 | 91.8 | 52.7 | 74.8 | 60.1 | 81.6 |
| | **LoRA-L+** | 56.6 | 89.5 | 47.3 | 67.9 | 69.0 | 70.5 |
| | **LoRSU-L+** | 63.0 | 91.2 | 54.2 | 74.7 | 76.7 | 82.1 |
| | **LoRSU-V** | 61.8 | 91.2 | 52.6 | 75.4 | 68.0 | 82.2 |
| **Offline** | **LoRA-L** | 59.0 | 91.1 | 55.1 | 75.0 | 65.0 | 81.2 |
| | **LoRSU-L** | 62.6 | 91.2 | 52.9 | 75.4 | 59.9 | 81.6 |
| | **LoRA-L+** | 59.5 | 91.1 | 51.7 | 72.7 | 72.0 | 75.7 |
| | **LoRSU-L+** | 62.0 | 90.8 | 54.2 | 74.6 | 75.6 | 80.7 |
| | **LoRSU-V** | 62.4 | 91.4 | 52.6 | 75.0 | 69.4 | 81.6 |

Table 17: Accuracy scores (%) for LLaVA. We fine-tune the LLM using LoRSU, and LoRA on the **CAn** dataset under different settings (the visual encoder remains frozen) and we compare its performance to our method LoRSU that fine-tunes the visual encoder (LoRSU-V). The suffix 'L' indicates that the method fine-tunes the LLM and 'L+' that the method fine tunes both the MLP projector and LLM.

| Setting | PEFT Method | VQA Datasets (Acc %) | | | | | |
| | | HM | DALLE | TSI | GTS | AIR | CAn |
|---|---|---|---|---|---|---|---|
| | **Zr-Shot** | 61.2 | 91.1 | 53.1 | 75.6 | 60.4 | 82.7 |
| **CL-5 shots** | **LoRA-L** | 60.0 | 90.9 | 53.0 | 75.2 | 59.3 | 80.5 |
| | **LoRSU-L** | 59.9 | 91.1 | 53.8 | 75.4 | 60.3 | 82.3 |
| | **LoRA-L+** | 60.8 | 91.2 | 49.0 | 74.5 | 60.2 | 83.2 |
| | **LoRSU-L+** | 60.8 | 91.5 | 49.7 | 74.0 | 59.6 | 83.6 |
| | **LoRSU-V** | 61.5 | 91.4 | 52.9 | 75.0 | 60.1 | 82.7 |
| **CL-20 shots** | **LoRA-L** | 60.0 | 91.7 | 51.5 | 73.8 | 55.2 | 71.5 |
| | **LoRSU-L** | 61.4 | 91.4 | 54.2 | 75.5 | 60.2 | 82.3 |
| | **LoRA-L+** | 59.5 | 92.0 | 48.5 | 72.1 | 56.9 | 81.7 |
| | **LoRSU-L+** | 60.9 | 91.2 | 48.1 | 72.8 | 57.9 | 82.2 |
| | **LoRSU** | 61.7 | 91.1 | 53.0 | 75.2 | 58.0 | 83.1 |
| **CL-50 shots** | **LoRA-L** | 59.8 | 91.7 | 53.0 | 73.7 | 55.4 | 69.5 |
| | **LoRSU-L** | 63.6 | 91.7 | 52.9 | 75.7 | 60.0 | 81.8 |
| | **LoRA-L+** | 60.0 | 91.5 | 40.4 | 68.9 | 54.2 | 69.8 |
| | **LoRSU-L+** | 64.7 | 89.2 | 46.7 | 71.6 | 55.0 | 72.7 |
| | **LoRSU-V** | 62.0 | 91.5 | 51.6 | 74.3 | 57.2 | 83.6 |
| **Offline** | **LoRA-L** | 61.2 | 90.9 | 53.1 | 74.3 | 58.9 | 79.1 |
| | **LoRSU-L** | 62.9 | 91.4 | 52.3 | 75.5 | 60.2 | 81.5 |
| | **LoRA-L+** | 61.5 | 91.2 | 48.5 | 72.8 | 59.4 | 84.6 |
| | **LoRSU-L+** | 63.9 | 90.9 | 49.1 | 72.5 | 60.2 | 83.3 |
| | **LoRSU-V** | 61.8 | 91.1 | 52.6 | 75.1 | 59.4 | 85.0 |

| Visual Encoder (Total #Params) | Method | Trainable #Params |
|---|---|---|
| **CLIP-L-14 (304.3M)** | LN | $0.1M$ |
| | F-FT | $304.3M$ |
| | F-EWC | $304.3M$ |
| | LoRA | $25.6M$ |
| | SPU | $20.0M$ |
| | LoRSU-Ours | $16.5M$ |

Table 19: Parameter efficiency for each method considered in our experiments.

Table 20: TFlops and time comparison between the fine-tuned LLM and the fine-tuned visual encoder using our method LoRSU (LoRSU-V) for the **offline** setting on **GTS** dataset. We report results based on a single NVidia A100 GPU. The table reports the same results as in Fig. 2.

| Method | Minutes/epoch | TFlops (Forward) | Trainable Params (M) | GTS Acc |
|---|---|---|---|---|
| **LoRA-L+** | 76.4 | 9.0 | 28.4 | 65.0 |
| **LoRSU-L+** | 78.6 | 9.1 | 21.6 | 75.6 |
| **LoRSU-V** | **5.5** | **0.36** | **16.5** | **91.1** |

## C PARAMETERS EFFICIENCY

Table 19 reports the number of parameters updated by each method and the percentage with respect to model size for both considered CLIP models. LN uses the least amount of parameters, however it lacks behind in accuracy on all evaluated datasets. LoRSU operates on fewer parameters compared to LoRa and SPU and yet strikes a strong balance between target datasets and the maintenance of generic knowledge, achieving the best performance in both classification and VQA tasks.

## D TSI DATASET CONSTRUCTION

To extract images from the videos of the Toyota Smart Home dataset (TSI), we discretized each video clip into 2 frames per second and then selected the frame in the middle of the total time duration of the video clip. In Table 23 we describe the actions that were selected and the corresponding prompt used for CLIP classification. We also note dropping few actions to avoid ambiguous classes.

## E EVALUATION OF CLIP ON XIMAGENET-12

In the Section 3 we evaluated CLIP robustness on XImageNet-12 benchmark Li et al. (2023b). Here we describe this experiment in more detail. XImageNet-12 benchmark Li et al. (2023b) covers 12 common categories from ImageNet and simulating six diverse out of distribution effects, such as overexposure, blurring, and color changing. Table 18 reports the results of CLIP ViT-B-16 with

Table 21: Ablation study on the influence of the rank for our **LoRSU** method with $k = 2$ top heads. We report the **last session** accuracy of CLIP. The model is fine-tuned on each session (5 in total) of **TSI** dataset using **50 shots** per session.

| Datasets | Zr-shot | Rank ($r$) | | | | | |
|---|---|---|---|---|---|---|---|
| | | 8 | 16 | 32 | 64 | 128 | 256 |
| **ImgNet** | 76.6 | 76.5 | 76.4 | 76.5 | 76.4 | 76.4 | 76.4 |
| **DALLE** | 90.9 | 93.3 | 91.8 | 91.2 | 92.3 | 91.8 | 93.2 |
| **TSI** | 13.2 | 56.0 | 55.5 | 56.6 | 58.0 | 56.3 | 57.1 |

Table 22: Ablation study on the influence of the rank for our **LoRSU** method. We report accuracy (%) scores for LLaVA using LoRSU with **top-2** attention heads. All fine-tuning methods use **TSI** data to fine-tune the visual encoder for **50 shots** in 5 sessions.

| Datasets | Zr-shot | Rank ($r$) | | | | | |
|---|---|---|---|---|---|---|---|
| | | 8 | 16 | 32 | 64 | 128 | 256 |
| **HM** | 61.2 | 62.1 | 61.9 | 61.4 | 61.9 | 61.6 | 61.6 |
| **DALLE** | 91.1 | 90.6 | 91.8 | 90.6 | 90.6 | 88.9 | 90.0 |
| **TSI** | 53.1 | 66.8 | 68.4 | 68.9 | 70.8 | 67.1 | 68.0 |
| **GTS** | 75.6 | 75.3 | 75.4 | 75.4 | 75.5 | 74.9 | 74.7 |
| **AIR** | 60.4 | 58.9 | 58.5 | 57.9 | 59.3 | 59.0 | 59.2 |
| **CAn** | 82.7 | 81.7 | 82.1 | 81.1 | 82.2 | 81.4 | 81.4 |

| Original Class name/Action | Generated Caption |
|---|---|
| Cook.Cleandishes | washing dishes |
| Cook.Cleanup | cleaning up |
| Cook.Cut | cutting food |
| Cook.Stir | stirring the pot |
| Cook.Usestove | ✗ |
| Cook.Cutbread | cutting bread |
| Drink.Frombottle | holding a bottle |
| Drink.Fromcan | holding a can |
| Drink.Fromcup | holding a cup |
| Drink.Fromglass | holding a glass |
| Eat.Attable | eating |
| Eat.Snack | ✗ |
| Enter | walking |
| Getup | ✗ |
| Laydown | lying down |
| Leave | walking |
| Makecoffee.Pourgrains | using a white coffee machine |
| Makecoffee.Pourwater | using a white coffee machine |
| Maketea.Boilwater | boiling water in a black kettle |
| Maketea.Boilwater | making tea |
| Maketea.Insertteabag | making tea |
| Pour.Frombottle | holding a bottle |
| Pour.Fromcan | holding a can |
| Pour.Fromkettle | holding a black kettle |
| Readbook | reading a book |
| Sitdown | sitting down |
| Takepills | ✗ |
| Uselaptop | using a laptop |
| Usetablet | using a tablet |
| Usetelephone | using a cordless phone |
| Walk | walking |
| WatchTV | watching TV |

Table 23: The original action names of the Toyota Smarthome dataset and their corresponding captions used to create the Toyota Smarthome Images (TSI) dataset. We use ✗ to denore the actions that are ambiguous and were not used to build the TSI dataset. The final prompt is created as "*The person in this image is {caption}*".

different pretraining. Although only one domain with random backgrounds of other objects exhibits weak performance, this could be attributed to model confusion between the two objects in the foreground and background, rather than a weakness in understanding the image.

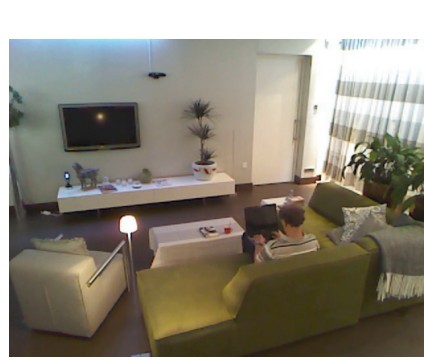 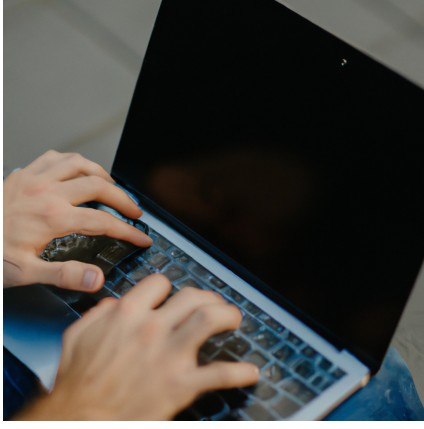

(a) TSI                         (b) DALLE

Figure 3: Use Laptop Example

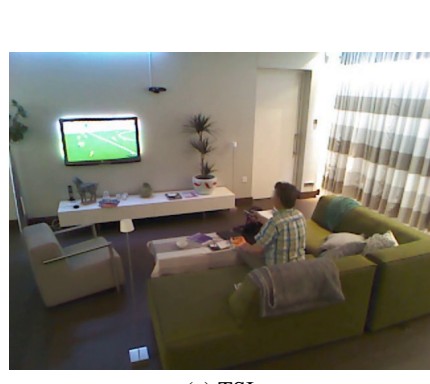 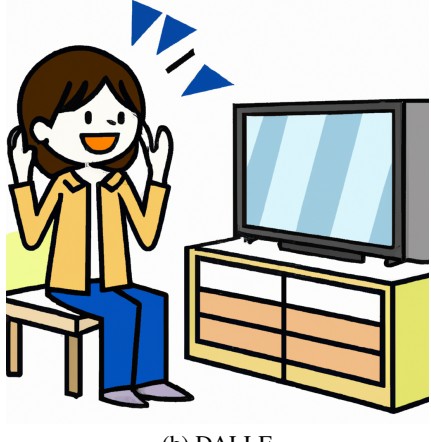

(a) TSI                         (b) DALLE

Figure 4: Watching TV Example

### E.1 EXAMPLES OF TSI AND DALLE DATASETS

We show additional examples of TSI images and DALLE generated images for some actions in Figures 3, 4, 5, 6, 7, 8, 9, 10, 11.

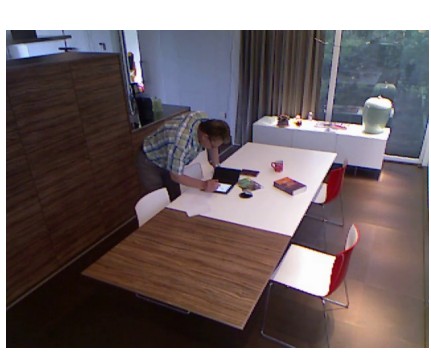 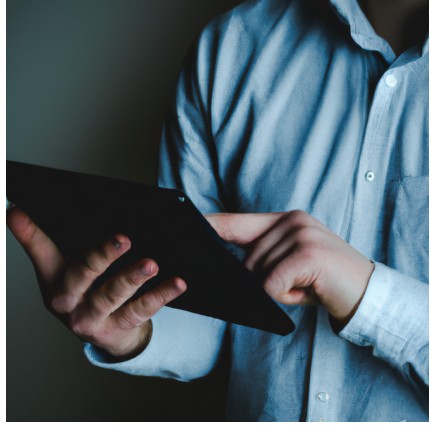

(a) TSI        (b) DALLE

Figure 5: Use Tablet Example

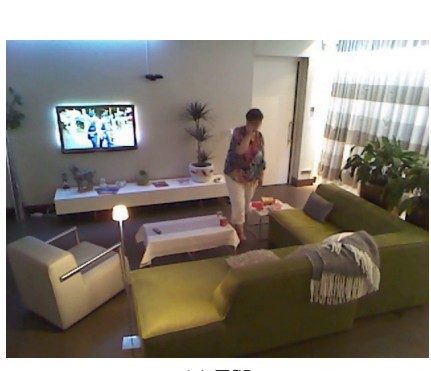 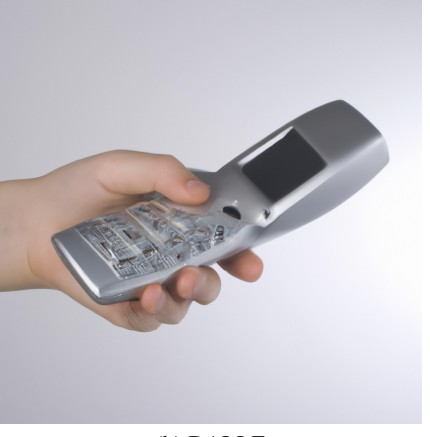

(a) TSI        (b) DALLE

Figure 6: Use a telephone Example

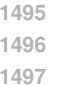 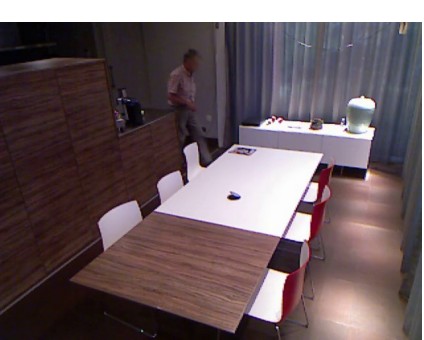 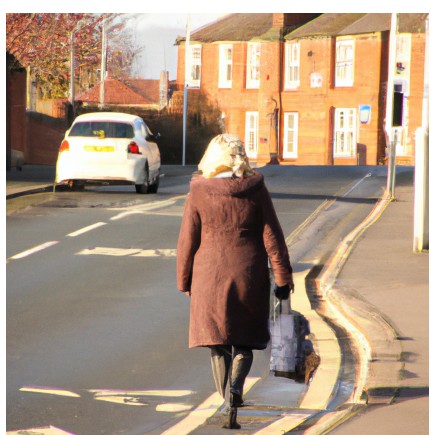

(a) TSI        (b) DALLE

Figure 7: Walking

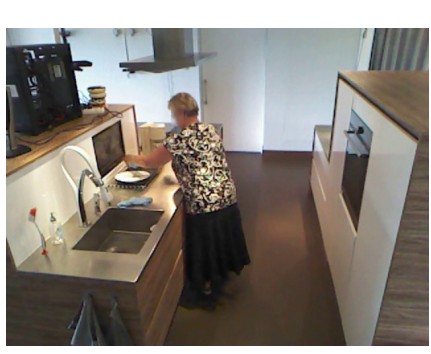 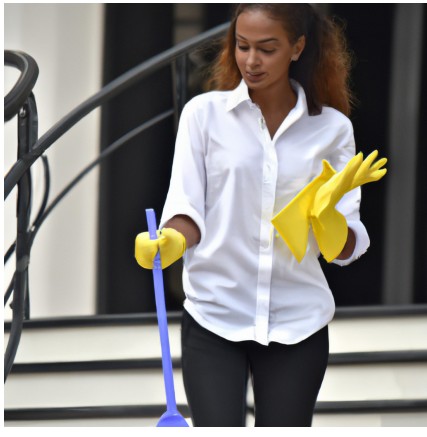

(a) TSI                                   (b) DALLE

Figure 8: Clean Up Example

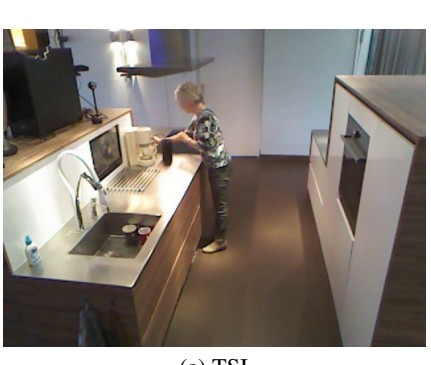 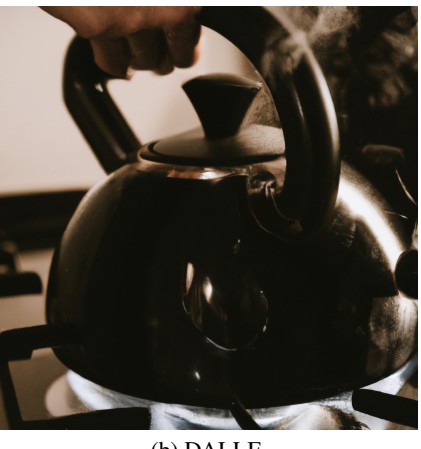

(a) TSI                                   (b) DALLE

Figure 9: Boiling Water in a Kettle Example

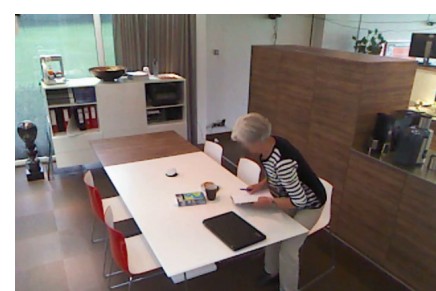 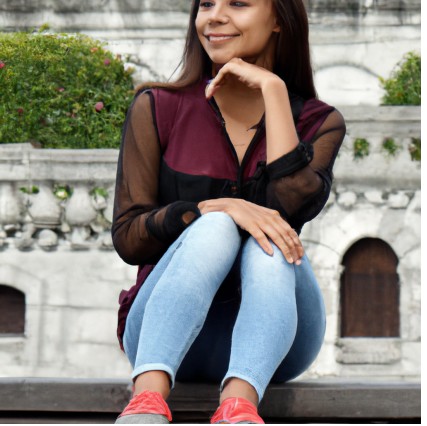

(a) TSI                                   (b) DALLE

Figure 10: Sit Down Example

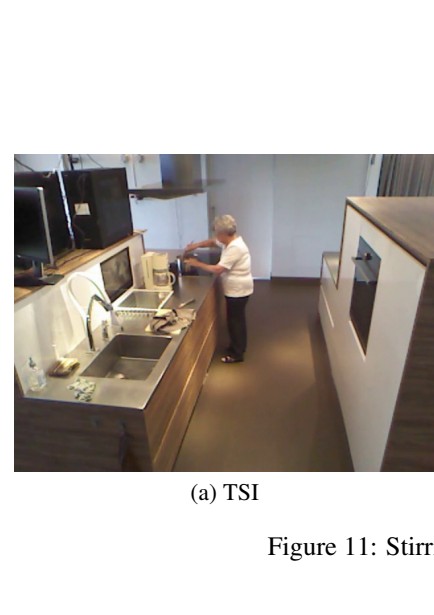
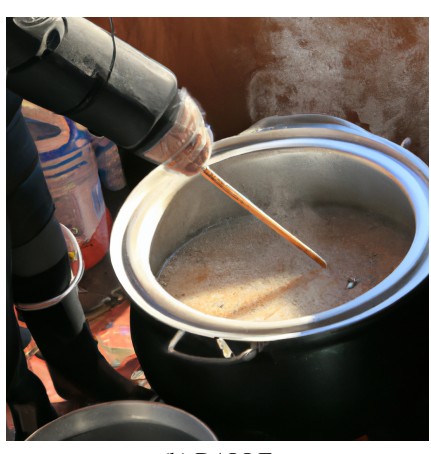

(a) TSI                                          (b) DALLE

Figure 11: Stirring The Pot Examlpe

