# OpenReview forum: "Adaptive Vision Encoders: Balancing Efficiency and Robustness in Vision-Language Models"
_ICLR.cc/2025/Conference — ICLR 2025 Conference Withdrawn Submission_

### Official Review · Reviewer_Mqpf · 2024-10-28

**Soundness:** 2
**Presentation:** 3
**Contribution:** 1
**Rating:** 3
**Confidence:** 3

**Summary:**

The paper shows the negative impact of CLIP encoder on downstream LLM, when the data is out-of-domain and hard for CLIP. The paper proposes a method that combines LoRA, SPU and selection of weights with large gradient to efficiently and effectively fine-tune the CLIP encoder, so that the problem can be mitigated.

**Strengths:**

1. The motivation is clear in that the paper demonstrates a clear case in which the CLIP encoder fails and also the downstream LLM.
2. The method seems to be highly efficient, especially when fine-tuning only the CLIP encoder without updating the LLM. It seems to give even higher performance with much less cost on computation.

**Weaknesses:**

1. The paper lacks novelty in the methodology. The proposed method is a straightforward combination of LoRA, SPU, and selection of weights (attention heads) with large gradient, without any interaction between these components. Meanwhile, the evidence of its effectiveness seems to be not strong enough, as is discussed in the following. The author might provide more evidence for the effectiveness of this straightforward method.
2. Some experimental evidence needs more justification:
    1. In Table 3 and also other tables, the difference between methods seems to be small (especially between SPU and LoRSU). The author might consider doing a significance analysis.
    2. In Figure 2 and table 13, it seems that just fine-tuning vision encoder is an efficient and effective strategy for LoRSU. However, the author should also include LoRA-V to have a fair comparison (also in Table 5). Also, the author should point out which version of LoRSU (L/L+/V) is used in the comparisons in Table 3 and other tables.
3. I personally think the section 4.1 is a bit redundant in that the proof is very straightforward and can be described just intuitively: since it basically tries to prove that selecting the largest components give us the biggest sum of components. The author might instead want to try to theoretically justify the deviate from the “local minima” (or solution space) given by the full gradient is small with the proposed method.

**Questions:**

1. The author might consider doing a significance analysis on tables that show the main comparison;
2. The author should include LoRA-V to the relevant comparisons.
3. The proof in section 4.1 is a bit redundant. The author might instead want to try to theoretically justify the deviate from the “local minima” (or solution space) given by the full gradient is small with the proposed method.

---

### Official Review · Reviewer_bHAs · 2024-11-01

**Soundness:** 2
**Presentation:** 1
**Contribution:** 2
**Rating:** 3
**Confidence:** 4

**Summary:**

The article tackles the problem of improving the performance of vision-language models across multiple visual domains and tasks. To achieve this, it proposes LoRSU (Low-rank adaptation with structured updates) that selectively updates a subset of the parameters in each transformer layer, i.e., the first linear layer of the MLP block, as in Zhang et al. (2024) and the most informative attention head (estimated via the task-specific loss). Experiments show that LoRSU achieves better or comparable results with existing adapters (e.g., LoRA, SPU).

**Strengths:**

1. Despite the overlap/findings present in other works, the analysis in Tab. 1 and 2, showing that CLIP issues propagate to LLaVA is a clear motivating example for the need to adapt the CLIP encoders.

2. The article combines existing techniques (i.e., SPU, LoRA) in a sound manner, achieving good results across a wide range of tasks.

**Weaknesses:**

1. Lines 112-128 are a repetition of lines 96-110. This unfortunate mistake harms the presentation because (i) it denotes a lack of thoroughness in proofreading the manuscript; (ii) it reduces the effective length of the paper to ~9.5 pages, taking out space from potential additional analyses or findings the reader may have benefited from. Presentation is part of the research process and carefully proofreading the manuscript is essential to present the contribution in the best possible way.

2. A core part of the manuscript is the presentation of the shortcomings of CLIP's visual encoder. This is stressed in lines 44-46 and verified on tests on action recognition datasets (Table 1): TSI (Toyota Smart Home) and a synthetic one (generated via DALL-E). There are two takeaways from these results: (i) CLIP has shortcomings on rare domains/distributions and (ii) those are propagated to large multimodal models using it (Table 2). This is considered a contribution of the manuscript, as stressed in lines 99-101 (and 115-117, due to the repetition). However, manuscripts have already shown the limitations of CLIP on benchmarks wider and more structured than the one presented in Table 1. Examples are [a,b,c] focusing on various types of compositionality (e.g., 8 textual modifications in [a], 10 challenges in [c]), [d] focusing on low resource challenges (i.e., rare domains) and [e] already showed how CLIP issues propagate to VLMs using it (Fig. 6 in their paper). The claim contribution (1) is not clear w.r.t. these works as well as the contribution of Sec. 3.

3. The technical contribution w.r.t. previous work is unclear. LoRSU combines two techniques: (i) selectively updates the parameters of the first linear blocks and (ii) selects which parameters to update based on the task-specific loss. The first has been presented in SPU, Zhang et al. (2024), as acknowledged in lines 92-93 and 225. Looking at the results presented in the appendix, SPU is often comparable to LoRSU, even outperforming it in some scenarios (e.g., Tables 7,9,10). For point (ii) the update is done via LoRA adapters, with the main difference being the focus on specific heads via the gradient of the task-specific loss (following what is done in SPU as well to select parameters). However, there is no ablation showing how the number of heads picked influences the final performance. All in all, there is a lack of analysis justifying the various design choices, with the advantages mostly shown via the empirical results on downstream datasets against the competitors. It would be helpful to include additional ablation studies/analyses (e.g., pruning ratio, where to apply LoRA layers etc.) and to expand on the contribution/practical advantages w.r.t. SPU and LoRA.

4. Following on the previous, the hyperparameters choice is not justified and thoroughly analyzed. For instance, Appendix B states that SPU and LoRSU use different sparsity ratios (e.g., 15% the first, 10% the latter) without analyzing the impact of this choice. The same goes for the data points used: appendix B states 800 data points to compute gradients, without further details on how they are picked.

5. It is hard to parse the results sa they are now. Instead of the most commonly used accuracy metric (adopted in Table 1 and Table 2), the main tables (3, 4, and 5) report target improvement and average control change. Those are harder to grasp, especially due to the lack of reference points to ground the results themselves. It would be better to report the results as done in the Appendix (i.e., with the natural accuracy choice) or use other metrics commonly used for continual learning (e.g., as done in SPU with average accuracy and forgetting). It could also be helpful to expand on why these metrics have been chosen (lines 355-364)/what they add w.r.t. those already present in the literature.

6. While it is always interesting to see methods linked to theoretical justification, the proof in Sec. 4.1 does not expand the principles previously defined. Specifically, Eq. (5) defines gradients as the criterion for pruning and Eq. (10) uses the same gradients to define the optimization problem, stating that we want to preserve only a subset of the heads (i.e., S). It simply follows that the optimal subset of the head is the one that leads to the largest overall gradient (i.e., the top-S). Note that this is simpler than a knapsack problem (stated in line 305) as there is no constraint on the capacity, just on the number of "items" to be selected. In the context of the proof, some elements are unclear (i.e., what does the intersection between $I_i$ and $I_j$ mean?) or not accurately defined (e.g., as per Eq. (5), $s_l$ is not bounded between [0,1], thus its sum could be greater than the number of layers in $I_l$).

7. The introduction heavily stresses the role of updating the vision encoder (contribution 2, lines 101-102 and 117-118). In Tables 5, 14, 15, and 16, the results are counter-intuitive as often the best results (or comparable) are achieved when tuning the language encoder (something that has already been studied in [f]). The analyses of the results in lines 480-497 also confirm the efficacy of updating the language side. This makes the message from the introduction and the experiments contradict each other: it would be better to clarify in the introduction that LoRSU is a general approach and that updating the vision encoder is not essential for achieving good results.

**Minors:**
- The claim "unseen domains" for CLIP (line 49) is hard to make as CLIP has been exposed to a huge amount of data and it might be exposed to virtually all domains but with different frequency. It would be better to replace "unseen" with "rare".

- Line 102 states that the method updates "the vision encoder [...] specifically on data where CLIP fails". This is slightly inaccurate, as the method does not take into account for errors of the model in the most common sense but rather takes a dataset as input (where CLIP potentially does not work well) and applies adapters there: the method per se has no notion of "data where CLIP fails" and it could be applied to any dataset given as input. This might be clarified.

- Line 146: LoRA is written incorrectly (it is not Low Rank Updates but Low Rank Adapters).

- Line 248: in the very last formula of the line, the subscript should be "k" and not "q" for W, as the gradient refers to the keys.

- Line 249: I could not find the definition of $\tilde{W}_o$.


**References:**

[a] Tristan Thrush et al., "Winoground: Probing Vision and Language Models for Visio-Linguistic Compositionality", CVPR 2022.

[b] Mert Yuksekgonul et al., "When and why vision-language models behave like bags-of-words, and what to do about it?", ICLR 2023.

[c] Cheng-Yu Hsieh et al., "SugarCrepe: Fixing Hackable Benchmarks for Vision-Language Compositionality", NeurIPS 2023.

[d] Yunhua Zhang et al., "Low-Resource Vision Challenges for Foundation Models", CVPR 2024.

[e] Shengbanc Tong et al., "Eyes Wide Shut? Exploring the Visual Shortcomings of Multimodal LLMs", CVPR 2024.

[f] Xiaohua Zhai, et al. "Lit: Zero-shot transfer with locked-image text tuning." CVPR 2022.

**Questions:**

Following on the weaknesses:
1. How will the presentation be improved?
2. How does the preliminary analysis (Sec. 3) advance the studies already present in the literature?
3. The are mixed results (e.g., SPU sometimes works comparably to LoRSU): when are the cases where the further LoRA fine-tuning of the most important heads is needed? and why?
4. What is the main technical contribution w.r.t. SPU and LoRA?
5. Is there a specific motivation behind focusing on the vision encoders tuning? Is it for the training/computational advantages shown in Fig. 2?
6. How have been the hyperparameters selected? and how the 800 datapoints used for importance estimation?

---

### Official Review · Reviewer_GrSw · 2024-11-03

**Soundness:** 2
**Presentation:** 2
**Contribution:** 2
**Rating:** 3
**Confidence:** 4

**Summary:**

The paper focuses on the OOD robustness of the present-day vlms. The authors hypothesize that the lack of robustness in the vlms stems from the vision backbone of the vlms. They propose ways to mitigate it with selective parameter optimization and test it on a continual learning and also offline setting. The results show improvements.

**Strengths:**

1. The studied problem is relevant. It is indeed helpful for improving the OOD robustness of the VLMs.
2. The method seems to provide some improvements on some benchmarks.

**Weaknesses:**

1. The writing lacks clarity. The paper can get a lot of help by improving the writing. For example, a paragraph is repeated 2 times in the introduction. Proofreading can be helpful.
2. Many details are missing in the paper. For example, what is DALLE in Table. 1? Table. 2 - how did the authors test these methods?
3. The paper also lacks suitable ablations to back the type of method chosen for the selection of parameters to update. What is the exact rationale behind it?
4. The writing is very dense in the evaluation section. After multiple readings, I cannot understand exactly the evaluated metrics. Also, what is the need to use these metrics? Is there some background literature on these? Specifically talking about Target Improvement and Average Control Change.
5. Simpler methods are not compared. Like peft methods. I can think of visual prompt tuning method from the top of my head.

**Questions:**

Most of the questions are listed in the weaknesses.

however, two additional questions:

1. what is the control set for the VQA?
2. why did the authors choose to present target improvement and not absolute results? the current state of presenting results is highly confusing.

---

### Official Review · Reviewer_FQnt · 2024-11-03

**Soundness:** 1
**Presentation:** 1
**Contribution:** 1
**Rating:** 1
**Confidence:** 3

**Summary:**

This paper presents a method to improve the robustness of vision-language models (VLLMs) on diverse domains. The proposed approach, called Low-Rank Adaptation with Structured Updates (LoRSU), selectively updates the model parameters to improve the robustness. Through experiments, the authors claim that LoRSU is able to improve the VLM performance with less than 10x compute. The author also provided theoretical justification on the strategy of proposed method.

**Strengths:**

N/A

**Weaknesses:**

Overall, this paper is not in good quality and is more like something generated by LLMs. Here are some of the reasons:
1. The argument of this paper are usuall unclear and meaningless (e.g. "DO WEAKNESSES IN CLIP PROPAGATE TO THE VLM" and "separately updating the vision encoder").  The content is usually out of context and not organized in a logical way.
2. The claims in abstract and introduction are not supported and inconsistent to context on the later pages (e.g., the claim "improvements on data where previous mistakes occurred" in abstract is never discussed in the method or experiment parts)
3. There are a lot of invented terms that are not consistent or do not exsit. For example, the datasets of TSI, DALL-E, GTS, AIR and CAn (Table1, 2 and 3), the model LLama-2+Pj and CLIP-L-14, and the method of LN, F-FT, F-EWC on Table 4.
4. The method and theory parts (section 4) do not make any sense.
5. The last 2 papergraphs of introduction (Line 95 and Line 111) are identicail.

There're also many other evidences that could be eazily identified in the paper.

**Questions:**

N/A

---

### Note · Authors · 2024-11-29

I have read and agree with the venue's withdrawal policy on behalf of myself and my co-authors.